# Graphene-driven correlated electronic states in one dimensional defects within WS$_2$

Antonio Rossi [1,2,3,11,13] ✉, John C. Thomas [1,3,12,13] ✉, Johannes T. Küchle[1,4], Elyse Barré[1], Zhuohang Yu [5,6], Da Zhou [7], Shalini Kumari [5,6], Hsin-Zon Tsai[8], Ed Wong[1], Chris Jozwiak [2], Aaron Bostwick [2], Joshua A. Robinson[5,6,7,9], Mauricio Terrones[5,6,7,9], Archana Raja [1,10], Adam Schwartzberg [1], D. Frank Ogletree [1], Jeffrey B. Neaton [3,8,10], Michael F. Crommie [3,8,10], Francesco Allegretti[4], Willi Auwärter [4], Eli Rotenberg [2] & Alexander Weber-Bargioni[1,3] ✉

Tomonaga-Luttinger liquid (TLL) behavior in one-dimensional systems has been predicted and shown to occur at semiconductor-to-metal transitions within two-dimensional materials. Reports of one-dimensional defects hosting a Fermi liquid or a TLL have suggested a dependence on the underlying substrate, however, unveiling the physical details of electronic contributions from the substrate require cross-correlative investigation. Here, we study TLL formation within defectively engineered WS$_2$ atop graphene, where band structure and the atomic environment is visualized with nano angle-resolved photoelectron spectroscopy, scanning tunneling microscopy and spectroscopy, and non-contact atomic force microscopy. Correlations between the local density of states and electronic band dispersion elucidated the electron transfer from graphene into a TLL hosted by one-dimensional metal (1DM) defects. It appears that the vertical heterostructure with graphene and the induced charge transfer from graphene into the 1DM is critical for the formation of a TLL.

One-dimensional (1D) systems in condensed matter physics provide unique insight into a variety of quasiparticle excitations, including charge density waves that arise due to Peierls instabilities[1,2], lossless transport through electronic wires in topological edge states[3-5], quantum spin liquids[6], as well as more exotic phenomena, such as Marjorana modes in nanowires[7] and the emergence of a Tomonaga-Luttinger liquid (TLL). The latter has been realized in both nanotubes and transition metal dichalcogenides (TMDs)[8-19]. These quasi particle excitations not only host new condensed matter physics phenomena but hold the promise to become major pillars of quantum electronics and quantum information applications[20]. We show the capability to controllably create 1D confined systems and to directly observe TLL

[1]Molecular Foundry, Lawrence Berkeley National Laboratory, Berkeley, CA, USA. [2]Advanced Light Source, Lawrence Berkeley National Laboratory, Berkeley, CA, USA. [3]Materials Sciences Division, Lawrence Berkeley National Laboratory, Berkeley, CA, USA. [4]Physics Department E20, TUM School of Natural Sciences, Technical University of Munich, Garching, Germany. [5]Department of Materials Science and Engineering, The Pennsylvania State University, University Park, PA, USA. [6]Center for Two-Dimensional and Layered Materials, The Pennsylvania State University, University Park, PA, USA. [7]Department of Physics, The Pennsylvania State University, University Park, PA, USA. [8]Department of Physics, University of California at Berkeley, Berkeley, CA, USA. [9]Department of Chemistry, The Pennsylvania State University, University Park, PA, USA. [10]Kavli Energy NanoSciences Institute, University of California Berkeley, Berkeley, CA, USA. [11]Present address: Center for Nanotechnology Innovation, Laboratorio NEST, Istituto Italiano di Tecnologia, Pisa, Italy. [12]Present address: Advanced Light Source, Lawrence Berkeley National Laboratory, Berkeley, CA, USA. [13]These authors contributed equally: Antonio Rossi, John C. Thomas. ✉e-mail: antonio.rossi@iit.it; jthomas@lbl.gov; afweber-bargioni@lbl.gov

formation at its native length scale, which is key to understanding the governing principles behind such a strongly-correlated system.

The hallmarks of a TLL are the independent dispersion of charge and spin, fractional charge transport, and power law suppression of the density of states (DOS) near the Fermi energy ($E_F$)[16–18,21]. The formation of a TLL was first observed in 1D carbon nanotubes[10,11], where the conductance of bundled single-walled carbon nanotubes showed power law scaling with respect to bias voltage and temperature. This has since been extended to a number of 1D systems such as semiconductor single-channel wires, nanowires, organic conductors, and fractional quantum Hall edge channels[6,10,15,17,21–27]. Recently, this phenomenon has also been shown to exist within 1D defects in two-dimensional (2D) TMDs at a plane of lattice points where the crystal structures on either side of the interface are mirrored, which defines a mirror twin boundary (MTB) within monolayer (ML) TMDs[16–18,28–31]. MTB formations have been predicted to exhibit metallic properties and to form out of sub-stoichiometric metal (M = Mo, W) or from depleted chalcogen ($X = S$, Se, Te) in $MX_2$ materials[24,32–35]. 1D chalcogen vacancy lines and Mo chains have also been shown to exhibit metallic character, where these defects have, in addition, been shown to form into or border other types of 1D defects through mass transport under thermal annealing conditions[28,30,31,36–38]. Each of these defects exhibit metallic behavior, and we now refer to these defects as 1D metals (1DMs), in an otherwise semiconducting TMD material. TLL formation in a 1DM within TMDs adds to the fascinating and vast array of material properties that 2D ML TMDs hold such as single photon emission, tunable band gaps, and strong spin-orbit coupling, to name a few[39–49]. Zhu et al. have applied scanning tunneling microscopy and scanning tunneling spectroscopy (STM/STS) to directly map out the local density of states (LDOS) associated with TLLs[18], where measurements show a gap opening near $E_F$, a length dependence on the highest occupied and lowest unoccupied state (HOS/LUS) band gap, and spin-charge dispersion observable in Fourier transform (FT) STS maps[16,17].

The influence of the substrate on TLL formation in 1DMs within a TMD is not yet fully understood, nor has it been systematically studied. Reports have shown that 1DM structures placed on Au and graphite show no evidence of TLL[17]. The hypothesis investigated in this report is that the vertical and direct contact of the 1DM on graphene is critical for the formation of a TLL. In this regard, a clear explanation is needed to connect the macroscopic band structure with the local electronic structure. In order to address these questions, we engineer 1DMs into 2D $WS_2$ grown on graphene with tunable control over both length and density via a post-synthesis, in-situ approach. Measurement techniques cross-correlating STM/STS together with spatially- and nano angle-resolved photo emission spectroscopy (nARPES) help to give a broad range of information both on the electronic band structure of the system and the nature of the defects[50]. Non-contact atomic force microscopy (ncAFM) further enables structure identification of 1DMs that host a TLL, where a metallic tip is functionalized with a CO molecule[34,51].

We report the role of the graphene substrate in TLL formation within TMD 1DM heterostructures. Two main types of 1DM defects are identified, in addition to combinations and strained subsets, where their controlled introduction enables TLL formation. Exclusive doping (charge transfer) from graphene into a 1DM gives on the order of ~3–8 electrons per measured 1DM, which brings the $E_F$ of graphene near the Dirac point, reducing its screening power and, therefore, increasing the electron-electron interaction in the 1DMs. We also observe an atomically localized band gap renormalization over the 1DM/TLL systems, with indications of the conduction band taking part in TLL formation. Additionally, as further proof that graphene is in fact playing a fundamental role and bolstering our hypothesis, when 1DMs are in contact with a wide band gap material, such as multilayered TMD atop graphene, the TLL structure does not appear. These findings underscore the unique role of graphene as an ideal substrate for TLL formation, striking a balance between charge donation and minimal screening, where, in comparison, metals over-screen electron-electron interactions and gapped systems lack sufficient free carriers. This insight opens new avenues for engineering 1D correlated states in 2D materials.

## Results

### Defect creation

In order to study 1DM defects that may host a TLL, we perform STM/STS studies on both unmodified $WS_2$, grown epitaxially via chemical vapor deposition (CVD) on a graphene/SiC substrate, and the same sample after $Ar^+$ sputtering and annealing in-situ to induce defectivity (Fig. 1a)[52–54]. Comparative results of low-energy sputtering in contrast to a low-temperature anneal are shown (Fig. 1b, c). As-grown samples are not significantly modified by annealing up to 250 °C, while chalcogen vacancies ($V_S$) start to form at 600 °C[47]. Low-energy $Ar^+$ sputtering at chalcogen creation sample temperatures ($SA_{step}$) greatly increases the density of $V_S$ and 1DM defects (see Supplementary Notes 1 and 2 and Supplementary Figs. 1 and 2 for both density calculations and SRIM simulations) compared to only annealing. Post $SA_{step}$ annealing at 600 °C ($SAA_{step}$) substantially reduces point defect density from the $SA_{step}$ [0.116 to 0.220 $V_S$/nm²] to the $SAA_{step}$ [0.001 to 0.007 $V_S$/nm²], where 1DM defects with increased length [$L_{SA_{step}} = 3.37 \pm 2.87$ nm, $L_{SAA_{step}} = 8.75 \pm 4.01$ nm] are formed (see Supplementary Fig. 1). Densities of 1DM defects are also reduced, to a lesser extent compared to $V_S$, from the $SA_{step}$ [0.012 to 0.030 defect/nm²] to the $SAA_{step}$ [0.008 to 0.010 defect/nm²], as 1DM defect length is increased via annealing. If multiple $SAA_{step}$ cycles are performed, longer 1DM defects [$L_{SAA_{3xstep}} = 12.95 \pm 5.01$ nm] can form (see Supplementary Fig. 3). After sputtering the sample for longer than 2 min, substantial degradation that is accompanied by an electronic structure change becomes measurable (see Supplementary Fig. 3e).

We make use of ncAFM with a functionalized CO tip to identify chalcogen vacancy lines, strained 1DMs resembling 4|4P or 4|4E intermediate defects, and combinations of chalcogen vacancies with strained 1DM defects (see Supplementary Figs. 4 and 5). These defects are characterized by their distinct electronic behaviors-most exhibit electron-like characteristics, while other 1DMs demonstrate hole-like behavior (see Supplementary Fig. 6). This classification is based on the modulation of electron density with bias[16,18]. Notably, the band dispersion observed in these defects mirrors that of known MTB structures[24,33,35]. For the hole-like behavior, our findings align with the 4|4E MTB, similar to what was reported by Jolie et al., and for the electron-like modulation, we observe this behavior in chalcogen vacancy lines and strained 1DMs resembling the 4|4P MTB found by Zhu et al. All structures display features of a TLL[16–18]. In addition to defect structures measured with ncAFM, we also show a power-law dependence in as-acquired STM images in single-line defects and fully formed 60° MTB formations, where this behavior (characteristic of a TLL) is not present in structures that instead contact an underlying layer of $WS_2$ (see Supplementary Fig. 7). An isolated 1D MTB in an otherwise unaltered $MX_2$ lattice would be highly strained. If isolated structures are formed, additional local strain-relieving features, such as Mo chains or chalcogen vacancies, can appear at either their endpoints or along their length, as demonstrated in Supplementary Fig. 5. This observation highlights a defect at the 1DM endpoint, where we measured a strained 4|4P intermediate symmetry thereafter. Our analysis reveals that among the types of 1DM defects measured, the isolated strained 1DM structures resemble intermediate defects in the process of forming fully relaxed MTB structures from $V_S$ defects[28,31,36]. Chalcogen vacancy lines, however, create less lattice strain of the isolated 1D defects measured with a single $SAA_{step}$ cycle, and likely represent the majority of defects created. While 1DMs have been more extensively studied in $MoS_2$ and $MoSe_2$, high formation energy in $WS_2$ has hindered atomic-scale investigations. We are able to take advantage of sulfur reduction techniques coupled with tandem atomic-scale

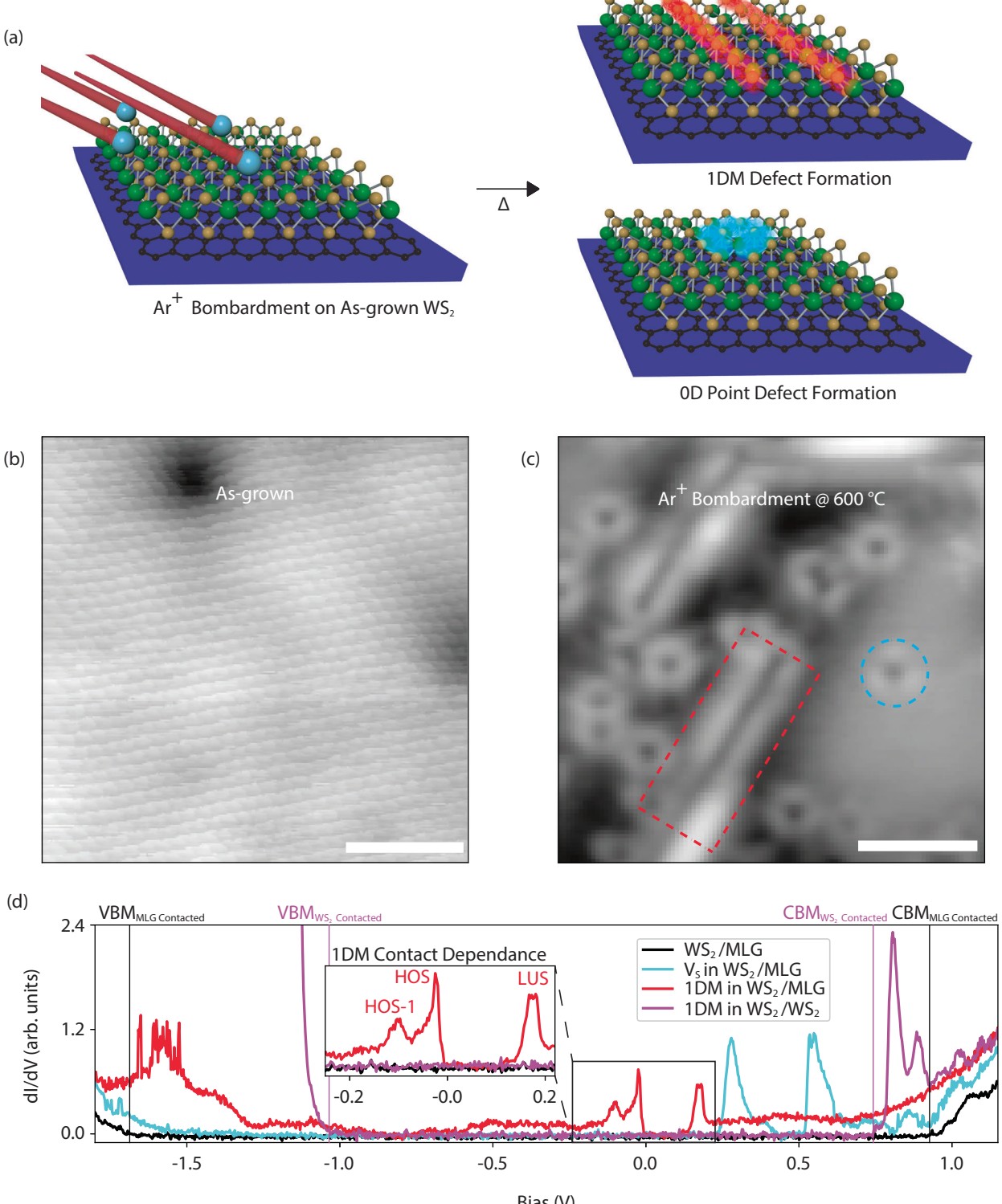

**Fig. 1 | WS$_2$ Defect Introduction. a** An overview of the defect creation process, where Ar$^+$ irradiation and annealing steps create both 0D and 1DM defects into an otherwise unmodified WS$_2$ monolayer. **b** Scanning tunneling micrograph depicting pristine WS$_2$ before Ar$^+$ bombardment ($I_{tunnel}$ = 30 pA, $V_{sample}$ = 1.2 V). **c** After the pristine sample is heated and exposed to an Ar$^+$ sputter ($I_{tunnel}$ = 30 pA, $V_{sample}$ = 1.2 V), both V$_S$ and 1DMs are present. Scale bars, 2 nm. **d** Point spectroscopy, in the form of the LDOS, for unmodified WS$_2$ in contact with monolayer graphene (MLG), V$_S$ within WS$_2$ and in contact with underlying MLG, a 1DM within WS$_2$ and in contact with MLG, and, for comparison, a 1DM within WS$_2$ instead contacted to underlying WS$_2$ (contact dependence is highlighted by the inset plot). States around E$_F$ for the MLG-contacted 1DM are labeled as HOS-1, HOS, and LUS. The valence band maximum (VBM) and conduction band minimum (CBM) are labeled with horizontal lines for unmodified WS$_2$ (black) and a 1DM in WS$_2$ contacted to underlying WS$_2$ (magenta).

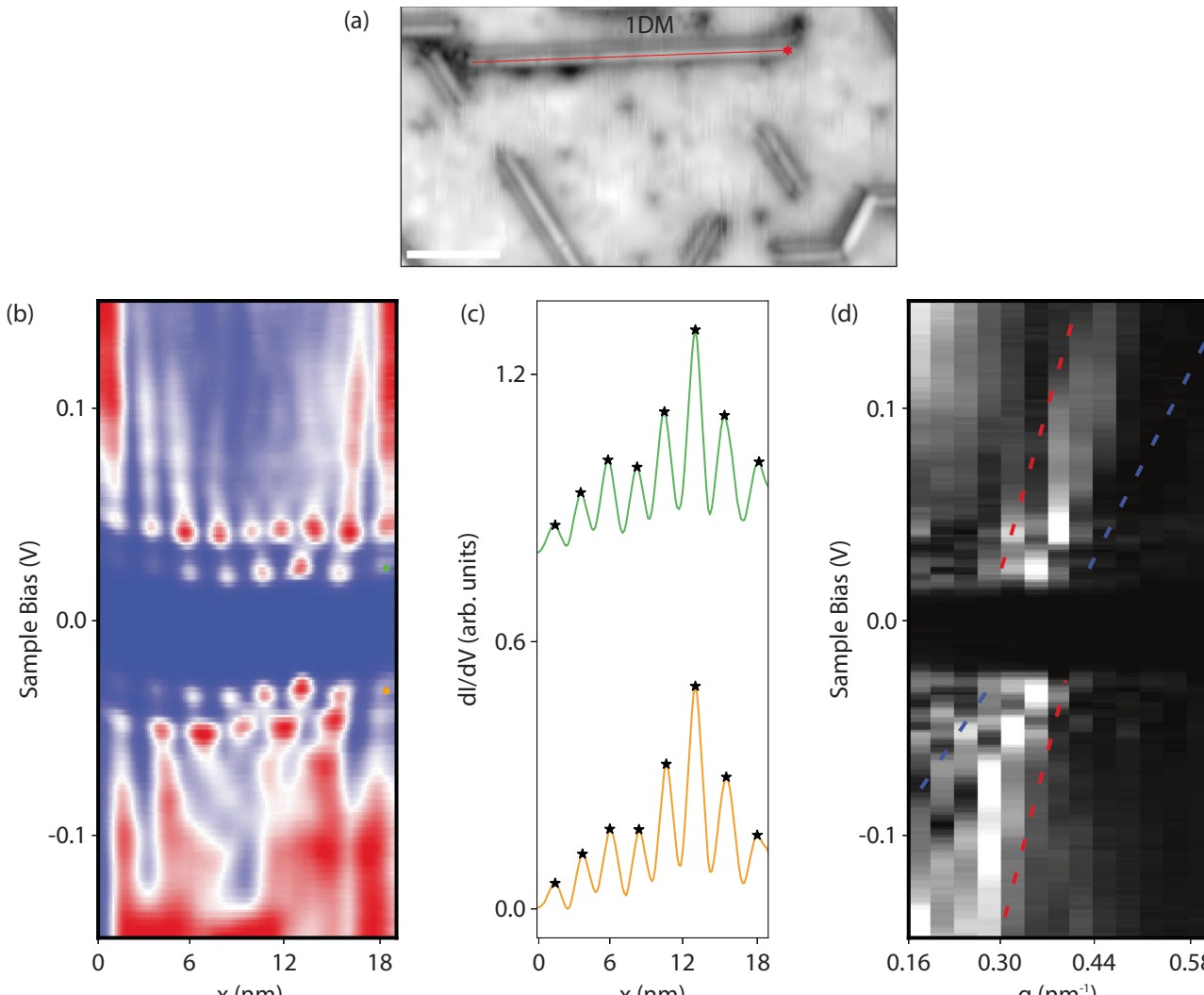

**Fig. 2 | Dense LDOS mapping. a** An electron-like 1DM is measured topographically, where the red line shown depicts a length of 18.96 nm ($I_{tunnel}$ = 20 pA, $V_{sample}$ = 1.4 V). Dense LDOS spectra is collected along (**a**) an electron-like 1DM (1 × 128 x 400 pixels) ($V_{modulation}$ = 5 mV, $I_{set}$ = 150 pA) beginning at the starred point along the red line. This is shown as a (**b**) function of bias and distance. The HOS and LUS are identified (**c**) at −0.032 V (orange) and 0.025 V (green). A 1D particle-in-a- box behavior is present above and below the HOS and LUS (maxima labeled as starred points). The number of nodes ($N$) increases linearly (N+1) from the LUS to LUS+1 (8 to 9), and also decreases from the HOS to HOS-1 (8 to 7). An FFT of (**b**) is shown in (**d**), where a spin and charge separation onset is seen above and below the $E_F$, and both the spin (blue) and charge (red) branches can be monitored from −0.15 V to 0.15 V. A $K_c$ value of 0.47 is extracted from (**d**).

measurements. Upon further investigation with STS, the two types of defects introduced during both $SA_{step}$ and $SAA_{step}$ are verified to be 1DMs and $V_S$, as detailed in Fig. 1d. Here, the band gap of surrounding $WS_2$ is measured to be on the order of 2.5 eV and $V_S$ shows deep unoccupied defect states split by spin-orbit coupled W $d$ states, which is consistent with previously measured values[47]. Conversely, 1DMs show an energy gap ($E_{gap}$) between the HOS and LUS around the $E_F$ that is much smaller compared to that of $WS_2$, and is dependent on 1DM length. An important finding is 1DMs engineered into bilayer and multilayer $WS_2$ (grown on the same graphene/SiC substrate) do not exhibit any sign of a TLL (shown in the spectra of Fig. 1d and Supplementary Fig. 7). This suggests a fundamental role of the substrate in the formation of a TLL. The HOS-LUS position around the $E_F$ can shift and is driven by, e.g., neighboring defects, tip-induced effects, and substrate variation[18,47]. Moreover, analogous to the discerned power-law behavior, no small gap is observed in defects formed on multilayer $WS_2$ (Fig. 1d). Evidently, one key ingredient for the formation of a TLL in 2D material heterostructures is the nearest vertical contact. This motivated the detailed study of the local and macroscopic electronic structure to unveil the mechanism behind TLL formation in $WS_2$ 1DMs that are in contact with graphene.

## Spatially dependent electronic structure

In order to better investigate the physics of the 1DM bandgap formed in $WS_2$, we make use of both point STS and differential conductance mapping, which are powerful tools for screening defects[34,47,55]. Figure 2 (a) showcases dense STS collected along an electron-like 1DM to extract behavior above and below the measured $E_{gap}$, where representative dispersion linescans of the HOS ($\psi_-$) and the LUS ($\psi_+$) are depicted in Fig. 2b, c. The number of quantum-well nodes changes as a function of bias, where the number of nodes increases in an integer fashion as the bias is ramped above and below the HOS and the LUS at −0.032 and 0.025 V in Fig. 2b, respectively. This 1D particle-in-a-box behavior, which has been demonstrated for both electron-like and hole-like 1DMs in $MoSe_2$ and $MoS_2$, shows an increasing number of nodes moving further past the LUS (with decreasing period) and a decreasing number of nodes moving below the HOS (with increased period) for the electron-like 1DM or the reverse behavior for the hole-

like 1DM[16,18]. This is consistent with our measurements shown in Supplementary Fig. 6 and contrasts electron-like behavior (the predominant 1DM defect measured) to hole-like behavior. We find that the nodal periodicity at the HOS and LUS for a measured in-phase electron-like 1DM structure (Fig. 2a) is 2.4 nm, and, for a 1DM showing out of phase behavior at the HOS and LUS (Supplementary Fig. 8), the periodicity at the HOS is near 3 and 2.4 nm at the LUS. These values are far above the lattice constant of 0.315 nm. We further outline the number of nodes as a function of length in Supplementary Fig. 8, where a single node is measured per $2.7 \pm 0.6$ nm defect length at the HOS. The measured periodicity for the inspected hole-like 1DM structure is 0.5 nm (Supplementary Fig. 6e–h) at both the HOS and the LUS, which is not within an integer relationship with the lattice parameter of $WS_2$. Supplementary Figs. 9 and 10 further show conductance maps as a function of energy below the HOS and above the LUS for both structures, where the presence of a neighboring $V_S$ scatters quantum-well states above the LUS and the number of observable nodes from the HOS to the LUS increases from 4 to 6 nodes, respectively, and this then increases from 6 at the LUS up to ten nodes at 0.4 eV before nearing the conduction band minimum (CBM) of $WS_2$ in Supplementary Fig. 9. The dense ($1 \times 128 \times 400$) linescan of point spectroscopy captured in Fig. 2b showcases expected spin and charge separation above and below the $E_{gap}$. As low-energy excitations are not Fermi liquid quasiparticles in TLL theory, the spin- and charge-density waves exhibit different dispersions and velocities, denoted as $v_s$ and $v_c$, respectively. Their ratio can be experimentally measured in the FT-STS measurement as $K_c = v_s/v_c$, where $K_c$ is the Luttinger parameter. From our FT-STS shown in Fig. 2d, we extract a $K_c$ of 0.47, which is near previously acquired values on other TMD systems[16,18]. $E_{gap}$ as a function of length is shown in Supplementary Fig. 11, where $E_{gap}$ is dependent upon length ($L$), scaling linearly with $L^{-1}$. This behavior stands in contrast to Peierls instability, where the $E_{gap}$ is constant in the CDW case and does not exhibit a length dependence[16]. We also look at the DOS measured in topographic STM images, as created defects are, on average, below the size regime shown to yield power-law scaling in point LDOS spectra[17,56]. TLL nodal oscillations are indistinguishable above the CBM of $WS_2$, however, a measure of signal intensity as a function of distance is feasible. Here, electron density is expected to decay according to $K_c$ as $\rho \sim x^{-K_c}$[17], where the DOS ($\rho$) measured under constant-current as a function of distance (x), in nanometers, is shown in Supplementary Fig. 7. Here, constant-current line profiles are extracted along multiple electron-like 1DM defects (sample bias = 1.2 V) which yield a power-law exponent value of $0.49 \pm 0.16$ that is near the value obtained in the FT-STS and in agreement with previously measured 1DMs that host a TLL[16–18]. We identify the TLL properties of these defects by observing a Luttinger parameter near 0.5, measuring an $E_{gap}$ opening near $E_F$, identifying an $E_{gap}$ dependence on 1DM length, visualizing 1D particle in a box behavior, and extracting evidence of spin-charge separation (See Supplementary Note 3). In previous studies, the observation of identical spectroscopic characteristics led to the determination of TLL within MTBs formed in other TMDs[16–18]. We summarize the spectroscopic characteristics of chalcogen vacancy lines, strained 1DM structures, and a fully formed MTB (formed across $WS_2$ edges) in Fig. 3. All of these defects exhibit metallic characteristics within $WS_2$ and the capability for hosting a TLL, which contributes to the local modification of the electronic DOS. The significance of the underlying contact in the electronic structure formation is discussed in the next section.

## Correlating angle-resolved spectroscopy and LDOS

We next perform nARPES to directly visualize the crystal band dispersion. The sub-μm probe in nARPES offers a spatial resolution capable of capturing the local inhomogeneity in the sample. Figure 4 showcases the as-measured band structure of the unmodified crystal versus the sample exposed to $Ar^+$ bombardment, where the sample was transferred from the STM chamber, after $SA_{step}$, in a nominally

inert environment to the nARPES chamber with an additional annealing step before nARPES acquisition. The two spectra are collected from the same sample, where a small region is found to be unaffected by the $SA_{step}$, due to sample holder shadowing during preparation. The spectrum obtained from the non-defective structure is displayed in Fig. 4a. It is collected along the Γ-K direction. Graphene and $WS_2$ keep epitaxial registry, therefore $WS_2$ also has the same crystal orientation[57]. A sketch of the two Brillouin zones (BZ) (graphene in black, $WS_2$ in green) is highlighted in the inset of Fig. 4a. Graphene exhibits sharp bands with the $E_F$ ~400 meV above the Dirac point. This is consistent with graphene prepared from thermal decomposition of SiC, where the carbon-rich buffer layer between graphene and SiC substrate creates an electric dipole at the interface affecting the chemical potential of graphene[58]. Such a native gating also affects the $WS_2$ band, whose top of the valence band maximum (VBM) appears to be ~1.48 eV below the $E_F$[57]. The local maximum at Γ appears below the maximum at K, confirming the ML nature of the TMD[59]. The introduction of 1DMs deeply affects the bands of $WS_2$. Fig. 4b shows the spectrum obtained from the region with high defect density. A substantial band gap renormalization is observed, where both the magnitude and chemical potential position are affected. The gap renormalization arises as the self-energies of the band-edge states shift due to the Coulomb interaction among free carriers[60]. The presence of 1DMs screens the Coulomb interaction in the $WS_2$ crystal leading to a smaller TMD gap. The HOS/LUS states, observed in STS, are not visible via nARPES experiment. The disorder dictated by the presence of defects results in a higher background signal and broader $WS_2$ bands. In order to have a clear signal of the HOS-LUS states, very ordered 1DMs with homogeneous length and orientation are necessary[23]. The $WS_2$ occupied electron bands are shifted upwards by ~500 meV. The position of the VBM is further confirmed by the spectrum collected with linear vertical polarization (see Supplementary Fig. 12). The graphene bands are also shifted up, although not by the same magnitude, with a subsequent change in the doping level. This is clear analyzing the Dirac bands collected near the K point of the graphene BZ along the direction highlighted in red within the inset of Fig. 5a, b. The momentum distribution curves, collected at the $E_F$ (Fig. 5c, d) can be fit with two Lorentzian functions. The position of their peaks defines a distance that approximates the diameter of the circle fitting the Fermi surface of graphene. Using Luttinger theorem[61–63], it is possible to extract the doping level being $n_p = 1.2 \times 10^{13}$ cm$^{-2}$ and $n_d = 2.5 \times 10^{12}$ cm$^{-2}$ for the as-grown and defective crystal, respectively (Fig. 5). The unmodified doping level agrees well with the value found in literature[61]. From these values and locally-measured defect densities (Supplementary Fig. 1 and Supplementary Note 2), we are able to determine that each 1DM hosts 3–8 electrons ($0.021 \pm 0.009 \frac{1DM}{nm^2}$). Our Monte Carlo calculations further suggest that our sample treatment does not have a direct impact on graphene (Supplementary Fig. 2). As a matter of fact, the presence of defects in graphene would open a gap at the Dirac point caused by an alteration of the system symmetry rather than shifting its chemical potential[64]. The shift of the Dirac point in graphene is the result of charge transfer from graphene to the newly formed 1DMs in $WS_2$. We can exclude a charge transfer involving the single $V_S$ having its characteristic in-gap state above Fermi level and therefore unoccupied, as displayed in the STS curve in Fig. 1d. Our findings suggest that graphene plays a significant role in the formation of a TLL, both donating charge to the newly formed defects and providing a weaker electronic screening due to the lower carrier density near the neutrality point. This, overall, increases the electron-electron interaction strength in the TMD.

An analysis of the energy distribution curves (EDC) intersecting the VBM is presented in Fig. 4d, denoted by the vertical dashed line in Fig. 4a, b. The EDCs are compared with STS curves obtained from the unmodified crystal and a 1DM formed on TMD. The pristine structure (shown as blue lines) exhibits a satisfactory alignment at the expected

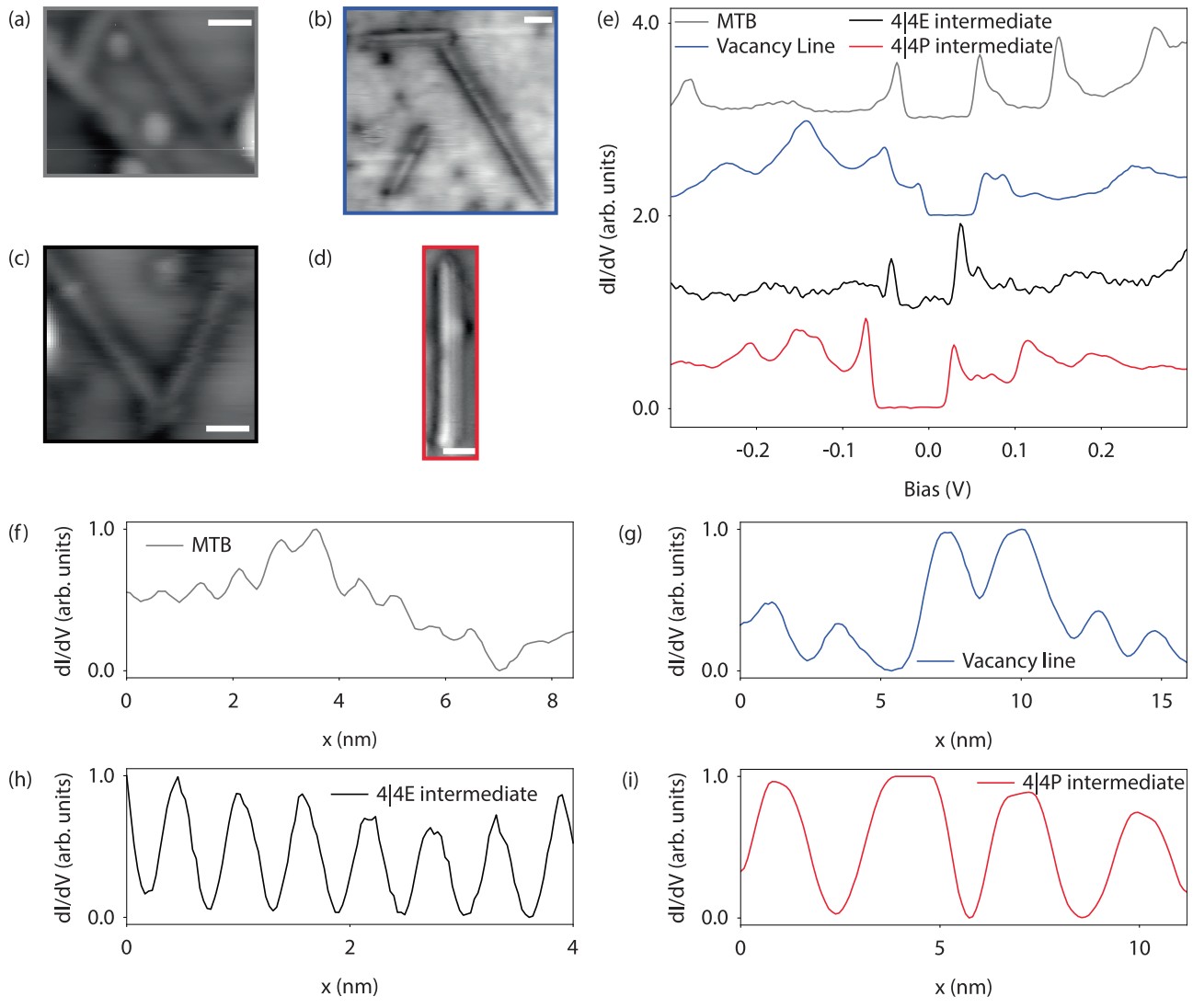

**Fig. 3 | Electronic structure comparison of different 1DMs.** Constant current image over **a** a fully relaxed MTB across WS$_2$ edge regions ($I_{tunnel}$ = 30 pA, $V_{sample}$ = 1.2 V), **b** a vacancy line structure showing 120 degree rotation within WS$_2$ ($I_{tunnel}$ = 20 pA, $V_{sample}$ = 1.4 V), **c** a 4|4E intermediate structure ($I_{tunnel}$ = 30 pA, $V_{sample}$ = 1.2 V), and **d** a 4|4P intermediate 1DM ($I_{tunnel}$ = 30 pA, $V_{sample}$ = 1.2 V). Scale bars, 2 nm. **e** dI/dV spectra recorded over each 1DM defect case, where all show a measurable E$_{gap}$ ($V_{modulation}$ = 5 mV) near the E$_F$. In addition, we highlight dI/dV linescans extracted from the HOS as a function of defect distance (**f-i**), which all exhibit similar oscillatory behavior.

onset of the VBM. The horizontal semitransparent bars, displayed in blue and red, highlight the energy region associated with the VBM onset for both the pristine and defective systems. A distinct kink in the EDC indicates the onset of the VBM as extracted from the nARPES data, marked by the black arrow. The observed band shift is primarily attributed to the band gap renormalization caused by the presence of the 1DM, in contrast to isolated V$_S$. Specifically, the onset of the VBM indicated by the STS curve obtained from the V$_S$ (represented in Fig. 1d) does not coincide with the onset of the VBM extracted from the EDC in Fig. 4b. However, the STS curve obtained from the 1DM (illustrated by the dark red line) exhibits the onset of the VBM that aligns with the EDC derived from the nARPES data acquired after the annealing step (depicted by the light red line). This behavior is additionally measured with spatially-localized STS acquisition in Fig. 4, where a shift in the VBM is seen approaching a 1DM with an STM tip. The measured onsets of the VBM (−1.16 ± 0.04 eV) and the CBM (0.74 ± 0.09 eV) were measured locally with STS, which corresponds with the appearance of the HOS and LUS. The relative shift of the VBM is +530 meV (CBM is shifted by −80 meV) and is depicted in Fig. 4c, which matches well to the band structure acquired by nARPES.

Spatially-resolved STS spectra reveal a band bending across the 1DM, akin to results by Murray et al.[65] and the pristine WS$_2$ that extends over 500 pm, giving rise to a sharp junction across the two regions. In addition, W 4$f$ and S 2$p$ core levels in Supplementary Fig. 13 show both core levels are rigidly shifted to lower binding energy in a similar fashion as the bands in the valence region. This suggests that the band gap renormalization is overall electrostatically driven. W and S peak fitting upon 1DM formation displays a weaker component that is further shifted toward a lower binding energy of an additional few hundreds meV due to the local chemical environment. Both nARPES measurements and STS highlight the importance in choice of substrate to determine the electronic properties of 1DM defects in TMDs. A screening effect is caused by graphene, which occurs due to the depletion of electrons in favor of 1DM states that enhances electron-electron interaction within the defect. This interaction leads to a metallic behavior that ultimately influences the renormalization of the band-gap in WS$_2$.

As anticipated above, graphene itself plays an important role in the formation of a TLL. In order to better understand its influence, we also examine the behavior of 1DMs in multilayered WS$_2$ on a graphene/

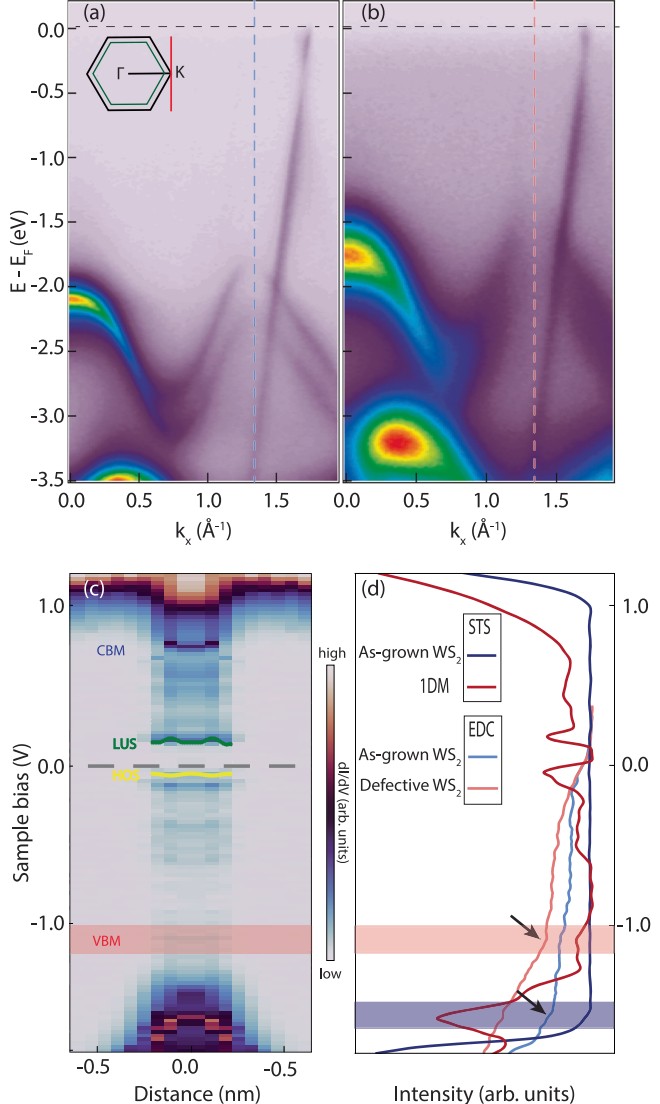

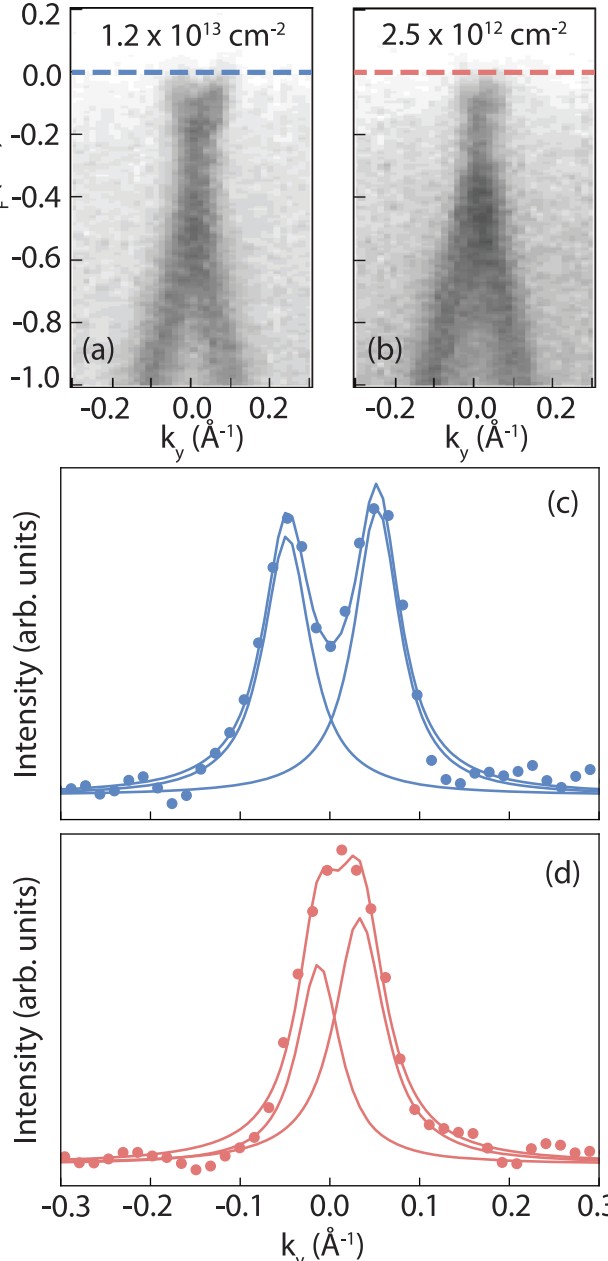

**Fig. 4 | nARPES band structure comparison. a** Unmodified and **b** defective band structure of $WS_2$ on graphene. The inset displays the $WS_2$ (green) and the graphene (black) BZ. The spectra are collected along the Γ-K orientation. **c** LDOS spectra collected from pristine $WS_2$ to a 1DM ($V_{modulation} = 5$ mV, $I_{set} = 150$ pA) and then recorded in reverse. The as-measured VBM, CBM, HOS, and LUS are highlighted with relative positions to $E_F$. **d** EDCs overlapped with STS from unmodified and defective $WS_2$. Dark blue and dark red are the STS signals collected on pristine $WS_2$ and on a 1DM, respectively. The light blue and light red lines are the EDCs collected at the K point of the BZ (dashed vertical lines in (**a** and **b**) respectively). Horizontal semitransparent bands display the onset of the VBM for pristine (blue) and defective (red), corresponding to EDC kinks clarified with black arrows.

**Fig. 5 | Graphene charge transfer.** The Dirac cone shift from the Graphene bands is related to the number charges transferred from the substrate into 1DM defective regions, where defective regions are closer to the neutrality point. nARPES spectra displaying graphene bands for both **a** unmodified and **b** defective samples. The spectra are collected along the portion of BZ highlighted by the red line of the inset in Fig. 4a. **c** and **d** display MDCs collected near $E_F$ for pristine and processed structure, respectively. The distance between the two Lorentzian peaks fitting each curve approximates the diameter of the Fermi surface, which is proportional to the carrier density in graphene. A smaller diameter indicates lower carrier density, highlighting the charge transfer process when $WS_2$ 1DMs are present.

SiC substrate (shown in Fig. 1d and Supplementary Figs. 7, 14, and 15) to compare direct $WS_2$/graphene contact against defects with greater separation from graphene. Defects shown in Supplementary Fig. 14a are over 3 ML of $WS_2$, which is confirmed by STS in Supplementary Fig. 14b and in agreement with similar measured $WS_2$ systems[66]. Dense STS is acquired over a 1DM in Supplementary Fig. 15, where there are consistently no in-gap states, or evidence of a TLL, across the entirety of the defect. Additionally, there are quantum-well states that form above the CBM of $WS_2$, which is in stark contrast with a 1DM in direct contact with graphene. As the electrons from the tip are tunneling into unoccupied 1DM states at positive sample bias with no reduced screening effects from the underlying substrate, the result is a 1D Fermi liquid. This behavior is further verified in constant-current line profiles

(Supplementary Fig. 7) over multiple defects within a multilayered $WS_2$ system. Contrary to 1DMs with direct contact to graphene, power-law fittings produce an exponential parameter of $0.99 \pm 0.15$ at a sample bias of 1.2 V. This has been shown previously with a 1DM over Au[17], where line profiles also decayed as a Fermi liquid. The presence of a 1D Fermi liquid in 1DMs with reduced proximity versus the presence of a TLL in 1DMs with direct graphene contact further highlights the importance of the underlying substrate. Cross-correlated

measurements made in this report are able to directly visualize the $E_F$ position related to the presence of 1DMs over graphene.

## Discussion

This work presents a new and controllable way to create 1DMs in $WS_2$ epitaxially grown on graphene. We show how 1DMs host a TLL, in which spectroscopic and topological signature is shown via STM/STS and ncAFM. A band gap opening is observed near the $E_F$ confirming the correlated nature of the electronic state inside the 1D defect. We demonstrate the formation of electron-like and hole-like 1DMs with combined STM/STS in $WS_2$ that display similar spectroscopic features but a distinct spatial and energetic difference in conductance image mapping. This behavior is due to respective electronic characteristics, which confirms earlier reports[16–18]. Data obtained with ncAFM paired with STM/STS also confirms the formation of a TLL within $WS_2$ fully formed MTBs, intermediate 1DM formations, and chalcogen vacancy lines. By means of nARPES, we were able to correlate scanning probe spectroscopic 1DM features to the band structure of the crystal. We observed how defective states behave as an acceptor of graphene electrons and that they cause a massive band gap renormalization of $WS_2$, where the presence of 1DMs screens the Coulomb interaction of $WS_2$ carriers, leading to an overall smaller TMD gap. The same effect is reflected on core levels, where we observe a similar chemical shift in binding energies. We also compare and contrast 1DM behavior directly over graphene to that of a 1DM with increased separation that is instead contacted to another layer of $WS_2$. Here, the effective electronic structure measured between STS and nARPES demonstrates that the direct heterostructure of graphene with a 1DM embedded into $WS_2$, and the subsequent induced charge transfer into the 1DM, is the driving mechanism behind the formation of TLLs and is critical in these types of systems.

The unique role of graphene is further highlighted by its ability to provide charge carriers while maintaining weak screening, a balance that is absent in either metallic or fully gapped substrates. Unlike metals, which over-screen electron-electron interactions, or semiconductors, which do not contribute sufficient free carriers, graphene's semi-metallic nature enables both charge transfer and the necessary electron-electron interactions for TLL formation. This demonstrates the fundamental importance of choosing the right substrate in engineering strongly correlated 1D electronic states.

Quasi-particle formation within 1D structures can have immediate relevance in quantum information processing, where application of such materials has yet to be manifested in functional devices. Additionally, electron transport across semiconductor to metal transitions may be beneficial in ultrafast electronic systems. Here, we provide a step-by-step approach to produce 1D metallic structures within a 2D semiconducting material, which holds relevance in atomic-scale, tailorable systems and electronic modification at the nanoscale. We further anticipate engineered TMD materials to be relevant in spin-polarized measurements, charge state effects, and spin transport.

## Methods

### Scanning probe microscopy measurements

All measurements were performed with a Createc GmbH scanning probe microscope operating under ultrahigh vacuum (pressure < $2 \times 10^{-10}$ mbar) at liquid helium temperatures ($T < 6$ K). Either etched tungsten or focused ion beam cut platinum iridium tips were used during acquisition. Tip apexes were further shaped by indentations into a gold substrate. STM images are taken in constant-current mode with a bias applied to the sample. STS measurements were recorded using a lock-in amplifier with a resonance frequency of 683 Hz and a modulation amplitude of 5 mV. FT-STS fit lines were derived from local extrema on spin and charge branches in accordance with analysis performed in Zhu et al[18]. A padding of ten pixels was used to showcase the region of separation.

In ncAFM measurements, a qPlus quartz-crystal cantilever was used (resonance frequency, $f_0 \approx 30$ kHz; spring constant, $k \approx 1800$ N/m; quality factor, $Q > 18,000$; and oscillation amplitude, $A \approx 1$ Å)[46]. The metallic tip was functionalized with a CO molecule for enhanced resolution[51].

### Angle-resolved photoemission spectroscopy measurements

ARPES experiments were performed in ultra high vacuum at $T = 6$ K at beamline 7.0.2 (MAESTRO) at the Advanced Light Source. The beam-spot size was $\approx 1$ μm. The photon energy for the valence band structure was $h\nu = 150$ eV and $h\nu = 350$ eV for core levels. The XPS curve-fitting analysis was performed using a convolution of Doniach-Sunjic and Gaussian line shapes superimposed on a background built of a constant, a linear component, and a step-function. For each S $2p$ spin-orbit doublet, a spin-orbit splitting of 1.2 eV and a branching ratio I($2p3/2$) : I($2p1/2$) = 2 : 1 (defined in terms of peak areas) were used. The W $4f$ spin-orbit doublets were fit using individual components with a spin-orbit splitting of 2.2 eV, to take into account non-linearities.

### Sample preparation

Islands of $WS_2$ were grown on graphene/SiC substrates with an ambient pressure CVD approach. An MLG/SiC(0001) substrate with 10 mg of $WO_3$ powder on top was placed at the center of a quartz tube, and 400 mg of sulfur powder was placed upstream. The furnace was heated to 900 °C and the sulfur powder was heated to 250 °C using a heating belt during synthesis. A carrier gas for process throughput was used (Ar gas at 100 sccm) and the growth time was 60 min. The CVD grown $WS_2$/MLG/SiC was further annealed in vacuo at 400 °C for 2 h.

$WS_2$ was sputtered with an argon ion gun (SPECS, IQE 11/35) that operated at 0.1 keV energy with 60° off-normal incidence at a pressure of $5 \times 10^{-6}$ mbar and held at 600 °C. A rough measure of current ($0.6 \times 10^{-6}$ A) enabled the argon ion flux to be estimated at ($1.5 \times 10^{13} \frac{ions}{cm^2s}$), where sample irradiation cycles spanned up to 30 s.

Samples were transferred from the STM to the nARPES chamber using an Ar (low-oxygen) suitcase to enable cross-correlative studies with minimal sample degradation risk. Samples were then annealed for 12 h at 250 °C and transferred to a 6 K sample stage for nARPES data acquisition.

## Data availability

All data needed to evaluate the conclusions exhibited are present in the paper and/or the supplementary information. The data generated for spatially-resolved STS (Fig. 2 (b)) in this study are provided in the Source Data file. Other data are available upon request. Source data are provided with this paper.

## Code availability

Software used for analysis are either presented in the supplementary information or can be provided upon request.

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

## Acknowledgements

The authors thank Alexander Stibor and John Turner, from the Molecular Foundry and the National Center of Electron Microscopy, and also Nino Hatter and Sebastian Baum, from CreaTec Fischer & Co. GmbH, for helpful discussions and experimental support. This material is based upon work supported by the U.S. Department of Energy, Office of Science, National Quantum Information Science Research Centers, Quantum Systems Accelerator. Additional support is acknowledged from the Center for Novel Pathways to Quantum Coherence in Materials, an Energy Frontier Research Center funded by the U.S. Department of Energy, Office of Science, Basic Energy Sciences. Work was performed at the Molecular Foundry and at the Advanced Light Source, which was supported by the Office of Science, Office of Basic Energy Sciences, of the U.S. Department of Energy under contract no. DE-AC02-05CH11231. J.C.T, A.R., and A.W.-B acknowledge support from the U.S. Department of Energy, Office of Science, Basic Energy Sciences in Quantum Information Science under Award Number DE-SC0022289. S.K and J.A.R. acknowledge support from the National Science Foundation Division of Materials Research (NSF-DMR) under awards 2002651 and 2011839. J.T.K and F.A. acknowledge financial support by the Deutsche Forschungsgemeinschaft (DFG) through the TUM International Graduate School of Science and Engineering (IGSSE), GSC 81.

## Author contributions

A.R., J.C.T., and A.W.-B. conceived and carried out the experiments. A.R., J.C.T., and J.T.K. carried out nARPES/XPS measurements with the assistance of C.J., A.B., and E.R. E.B., and J.C.T. contributed to SRIM/TRIM simulations. A.R., J.C.T., and J.T.K. carried out ncAFM measurements with the assistance of H.-Z.T. and M.F.C. A.R., J.C.T., and J.T.K. performed all STM/STS experiments with additional contributions from A.R., E.W., A.S., D.F.O., F.A., W.A., and A.W.-B. A.R. and J.C.T. performed all nARPES related data analysis with support from C.J., A.B., and E.R. A.R and J.C.T. performed all ncAFM, STM, and STS-related data analysis with support from A.R., E.W., A.S., D.F.O., J.B.N., M.F.C., F.A., W.A., A.W.-B. Z.Y., T.Z., S.K., J.A.R., and M.T. synthesized the samples. All authors discussed the results and contributed toward the manuscript.

## Competing interests

The authors declare no competing interests.
