## [Transparent Peer Review file · Nature Communications]

Graphene-Driven Correlated Electronic States in One Dimensional Defects within WS₂

Corresponding Author: Dr John Thomas

Version 0:

Reviewer comments:

Reviewer #1

(Remarks to the Author)

I read the paper with great interest, the authors investigated the WS₂'s MTB formed on graphene, which shows TLL properties. The form of TLL nature is related to the charge transfer between MTB and the Graphene. The work has its significance in showing us the influence from the substrate in formation of TLL, however, it's not clear (at least to me) that the gap renormalization is due to 1D correlation in MTB or due to MTB's other influence. Other comments is listed as follows

1. Fig.1 show the as-grown WS₂ and the one sputtered by Ar ions. I guess the authors want to show us the comparison between the two samples, but the figures are shown with different scale bar, making the comparison less intuitive. Besides, the Fig.1b is not enough clear, although I could identify the 1D TLL and the vacancy, I still suggests the authors could add some marks on the figure.

2. The spin-charge separation in E-K relation revealed by STM is not easy to identify in SFig.8 (spin's E-K), especially compared to Feng Wang's results (Nat Mater, 21, 748).

3. The correlation gap near E_f in MTB is highly unsymmetrical about E_f , E_f is almost at the edge of HOS, why

4. The shift of E_f in Graphen is attributed to the transfer charge to MTB, but not charge transfered from graphene to the Vs. Is this because the unoccupied states for Vs locate at much higher level?

5. Can the correlated gap observed by STM be fitted with power-law function in TLL model.

6. some typos, like in first line in discussion part

7. Discussion part should include more discussions about mechanism that how transfer charge between Graphene and MTB induce the gap renormalization.

8. Some other works relating to TMD's edge state should be referred

Proc. Nat. Acad. Sci. USA 113, 8583–8588 (2016)

Nat. Commun 11, 659 (2020)

Reviewer #2

(Remarks to the Author)

The authors study electronic structure of mirror twin boundaries (MTB) and Tomonaga-Luttinger liquid (TLL) behavior in particular, in WS₂ using scanning probe microscopy and nARPES. MTBs are introduced to WS₂ by Ar-ion bombardment and annealing. The authors claim to elucidate on how the substrate affects TLL and find charge transfer from graphene to MTB, which in turn leads to band shifts.

In my opinion, the insights gained onto electronic structure and TLL of MTBs in this manuscript are not new and the presented results are not of particularly high quality (compared to those already reported in the literature). Ar-ion bombardment induced MTB formation and nARPES data of such samples might be new, but not enough to warrant publication in Nature Communications.

In more detail:

1. TLL is not conclusively demonstrated.

The change in the periodicity isn't particularly evident in Figure 2 or SI Figs 5 and 6.

"We also highlight nodal increase as a function of bias in Fig. 2 (c), where the number of nodes moves from 6 at the LUS up to 10 nodes at 0.4 eV before nearing the CBM of WS 2 for the SMTB."

Here, counting of nodes should refer to the nodes in the quantum well states, and the size of each node should remain about the same. In Figure 2(c), the size of the nodes seem to vary (if counted to have e.g. 6 nodes at LUS), apparently due to overlap of the atomic features and the quantum well states. It is then not clear how to find the number of QW-state nodes. In Figure 2(f), the sizes are consistent, but then the change in the number of nodes is only one:

"This is also shown for an AMGB in Fig. 2(f), where 8 nodes are measured at the HOS and 9 nodes are measured at -0.3 eV."

In SI Fig 5 it is difficult see how many nodes is there at each bias voltage. In SI Fig 6 the number of nodes changed from 6 at $V=600$ mV to 8 at $V=-250$ mV, and difficult to read outside this range. The overall picture is easier to grasp from Figure S7, but the quality is mediocre. The spin- and charge-band dispersions in Fig. S7(c) are a leap of faith, as is the DOS fit in Figure S9.

The presentation is lacking in places. Figure 2, it is not clear if the whole MTB is shown or just a small part of it? In (c) and (d), it is not clear what is shown, is it a collection of conductance maps or is x-axis position coordinate along the MTB and y-axis bias voltage?

2. All of those results related to TLL have already been presented in the literature, but in higher quality with all these features clearly resolved, e.g., in Ref. 16 where also the graphene doping is studied via gating.

The charge transfer from graphene to the 2D materials is discussed (in more detail) in, e.g., C. Murray et al., ACS Nano 14, 9176 (2020). In fact, the authors do not seem to discuss the band banding issue here at all, but simply assume that the charge transfer from graphene to defective WS2 yields a homogeneous band shift, which I would think is incorrect.

3. The authors don't obtain insight on the role of substrate on the electronic structures. The authors only compare samples without MTBs to samples with MTBs on the same substrate and thus it is difficult to draw conclusions on how the substrate affects MTBs. For that there should be a comparison between different substrates.

Moreover, there is no data on how the TLL is affected by doping and while the defective samples contain both vacancies and MTBs, their relative roles to the doping (and TLL) cannot be separated. As a minor note, the authors write that MTBs with increased length are formed in SAA_step (as compared to SA_step), but why the Table in Suppl. Fig. 1 shows decreased defect density for SAA_step.

Reviewer #3

(Remarks to the Author)

This manuscript reports an interesting method to create mirror twin boundaries (MTBs) in WS₂ monolayers. As I learned from the manuscript that the large formation energy for S deficient structures in WS₂ hinders the formation of MTBs in it. The authors used low-energy Ar⁺ sputtering to create S vacancies for substantially reducing the S concentration in WS₂ and then annealed the sample to form MTB defects, which promote the formation MTBs. Given the prepared MTB structures, the authors found that charge transfer from the graphene substrate, providing weak electronic screening, enhances electron-electron interactions in MTBs and thus forms TLL states within 1D MTBs. Although the preparation of MTBs in WS₂ seems novel to me, observations of TLL states in MTBs of other TMDs were previously reported (e.g. DOI: 10.1103/PhysRevX.9.011055, 10.1021/acsnano.0c05397), also from the group(s) of some of the authors. In light of this, I was not fully convinced that this manuscript is justified for publication in Nature Communications, especially in the absence of discussion on the growth process and mechanism. My detailed comments are as follows.

1. As the two-step preparation protocol is novel for building MTBs in WS₂, I strongly urge the authors provide more information on the growth mechanism. I appreciate that that the authors carried simulations on the creation of S vacancies, but it is unclear that the kinetic process and the formation energy of S vacancies transforming into MTBs.

2. In Fig.1(e), there is another peak sitting at the right side of the HOS. does it arises from a folded state of HOS or represents another band residing below HOS? Please the authors clarify.

3. I was curious why both the HOS and LUS were observable in STS solely, but were invisible in ARPES measurements, even at 6 K. Could the authors explain?

4. All minus signs in the E-E_F label of Fig. 3(a) are missing; Those two shadowed bars in Fig. 3g were not explained in the associated caption and/or the main text.

5. The authors compared linecuts of ARPES maps with STS spectra in Fig. 3(g) and claimed that they are well consistent. With due respect, I was not convinced that they are "consistent" from any energy range plotted in Fig. 3(g). Please the authors explain, by comparing which characteristic(s), they are consistent. In addition, I noticed that the linecuts of the

ARPES maps hit the energy axis above E_F . Could the authors explain this?

6. The main text, and the SM, does not mention what information Fig. 4(b) provides. It looks like a TOC figure, but it was, perhaps, placed in a wrong place. The authors should either explain it or remove it from Fig. 4.

Version 1:

Reviewer comments:

Reviewer #1

(Remarks to the Author)

The author addressed my concerns carefully, I think the work is interesting enough for publication.

Reviewer #2

(Remarks to the Author)

The authors have revised the manuscript and provided detailed response to the Reviewers' criticism. Some of the raised issues were satisfactorily addressed, some were not.

However, I will not go through these in detail, since I realized that the manuscript contains a major flaw: Vast majority of the defects shown in the manuscript are not mirror twin boundaries (MTBs)! E.g, those in Figs. 1(b), 2(a), and 5(a) are all short segments that begin and end from pristine WS₂, which is impossible for MTBs since the lattice orientation must be mirrored in the two sides. In fact, Figs. 1(b) and 1(c) both look like the lattice orientation is not mirrored (in Fig. 1(a) I am looking at the orientation of the 3-fold symmetric vacancy features). The only case that could be MTB is in the first panel of Supplementary Fig. 9.

These line defects are instead likely vacancy lines, which would also agree with the previous work from MoS₂. Observing quantum well states and strong electronic correlations in vacancy lines could be novel. However, in order to address this issue and possible changing the story to vacancy lines, everything in the manuscript has to be changed and therefore the present manuscript cannot be considered for publication in Nature Communications.

Reviewer #3

(Remarks to the Author)

I've carefully read the authors' reply and the revised manuscript. I appreciate with the authors' efforts for improving it. The authors managed to clarify most of my concerns and to strengthen the importance and significance of the manuscript. I thus recommend it for publication in Nature Communications.

Version 2:

Reviewer comments:

Reviewer #2

(Remarks to the Author)

Below are my previous report (paragraphs starting with >>), the authors' response (starting with >), and my response (starting with .).

>> The authors have revised the manuscript and provided detailed response to the Reviewers' criticism.

>> Some of the raised issues were satisfactorily addressed, some were not. However, I will not go through these in detail, since I realized that the manuscript contains a major flaw: Vast majority of the defects shown in the manuscript are not mirror twin boundaries (MTBs)! E.g, those in Figs. 1(b), 2(a), and 5(a) are all short segments that begin and end from pristine WS₂, which is impossible for MTBs since the lattice orientation must be mirrored in the two sides. In fact, Figs. 1(b) and 1(c) both look like the lattice orientation is not mirrored (in Fig. 1(a) I am looking at the orientation of the 3-fold symmetric vacancy features). The only case that could be MTB is in the first panel of Supplementary Fig. 9.

>> These line defects are instead likely vacancy lines, which would also agree with the previous work from MoS₂. Observing quantum well states and strong electronic correlations in vacancy lines could be novel. However, in order to address this issue and possible changing the story to vacancy lines, everything in the manuscript has to be changed and therefore the present manuscript cannot be considered for publication in Nature Communications.

> We greatly appreciate your thorough evaluation of our revised manuscript and your constructive feedback. The main criticism of the reviewer is that the observed line defects cannot be mirror twin boundaries because of the lack of full lattice orientation inversion. We are wondering if the issue is partly semantics, because we clearly demonstrate inversion symmetry associated with MTBs and the lack of chalcogen vacancies within the 1D defect line using ncAFM. In addition, we show

clear spectroscopic signature of TLLs that have been previously shown for both types of MTB symmetries we present within WS₂. We do realize that MTBs in TMDs are typically associated with large 1D features (linear or triangular) that evoke an inversion symmetry across the entire flake, which are typically introduced during the growth process. We are demonstrating a new way to engineer these 1D defect lines post growth which exhibit the exact same inversion symmetry as MTBs, including the formation of a correlated electron system, in this case, a TLL. However, to be more clear, in terms of semantics and to avoid any confusion, we now refer to 1D inversion domains (IDs).

. No, there is no semantics. It is simply impossible to have finite-length 1D defect with inversion symmetry on the two sides. Where is the inversion symmetry disappearing at the end of the 1D defect? The lattice cannot transform from “normal” to “inverted” orientation without defects. Simply try to create an atomic model for a finite length MTB segment (based on the infinite MTB structure shown in Fig. 4) and you will quickly realize this.

. Because of this it is also obvious why all previous research report MTBs “that evoke an inversion symmetry across the entire flake” (or large triangular domains). It’s not because the growth process happens to lead to long MTBs between inverted domains, but because that’s the only way to connect inverted domains; by going all the way from edge to edge.

> The main finding and critical message of this manuscript is that we show that 1D IDs are not sufficient to exhibit a TLL phase alone, and that it requires a heterostructure with graphene. We don’t observe the formation of a TLL in 1D IDs when it is contacted to an underlying WS₂ layer. We realize that to convince the readers that our finding directly applies also to TLLs found in MTBs, we need to provide a clear description that shows the equivalency in terms of the structure of the 1D IDs with a traditional MTBs. We are providing a brief discussion of the main arguments to the reviewer here, which we also argue in the updated manuscript.

. That may be true for whatever this defect is, but it is not MTB (nor 1D ID).

> First and foremost, we employ, parallel to scanning tunneling spectroscopy (STS), ncAFM that provides precise insight into the local atomic structure of the feature we are observing (see updated Supplementary Fig. 4). We observe two inversion symmetries of the three that are predicted to be the most stable in the formation of 1D defects in TMDs^{1–4}. In our case we observe the 4|4E and 4|4P inversion symmetry (referred to AID and SID). These two types of inversion symmetries that we find experimentally are the ones found in MTBs that exhibit TLLs. We show that the electronic structure we measure via STS (shown in Supplementary Fig. 5 below) is what has been observed and predicted for both the purely 1D and 1D 60° 4|4E structures⁵ and consistent with measurements for both the purely 1D and 1D 60° 4|4P structure^{6–8}. A ncAFM methodology for atomic species identification, described by Barja et al.⁶, is used identify both types of IDs.

. First, in many of the STM/ncAFM images shown in the manuscript, the image has been cropped in such a way that it is impossible for me to judge whether it is a true MTB or a finite-length 1D defect. Some of these defects might be one and some might be the other.

. Second, while the structure in Supplementary Fig. 4 indeed looks similar to MTBs reported in the literature, it does not show inversion symmetry. Looking at the yellow and green dots, the lattice orientation is the same, and the red triangle is drawn wrong (the right side is missing yellow dot that would fall inside the triangle).

> We do agree with the reviewer that 1D vacancy lines would provide a fascinating playground, e.g., to study the hopping mechanism that could lead to 1D transport. However, not only have we clearly demonstrated the inversion symmetry with ncAFM and that we do not measure the presence of chalcogen vacancies, we have extended experience creating 0D and 1D defects in these systems, but we have so far not been able to align more than 4 vacancies in a true 1D line regardless of the formation mechanism (UHV annealing, laser annealing, He⁺ ion bombardment, Ar⁺ ion bombardment) and subsequent annealing protocols. Moreover, our 1D IDs are between 2–17 nm.

. To be clear, I am not claiming that it is 1D vacancy line. It was just the first candidate that came to mind based on previous literature.

> We believe based on the reviewers doubts, it is important to point out that our approach to create these MTBs goes through an evolution. At first we create dense regions of both chalcogen vacancies and 1D IDs. Upon additional annealing, we find longer and longer 1D ID formations that start creating lattice-oriented zig-zag networks (as shown in Fig. 1 (b) and Supplementary Fig. 1) with a large reduction of chalcogen vacancies.

. And these I would agree look like MTBs since each domain appear to be bounded by MTBs from all sides, and MTBs are connected by local defective regions.

> Similar mechanisms were observed in TEM studies of creating 1D defects, which are introduced under ebeam exposure and immediately start to form into either MTBs or IDs^{2,9,10}. Something that is consistent across much of the literature is that elongated 1D vacancy lines form 1D IDs by means of mass transport⁹. We are glad the reviewer brought this up, as we do not want to confuse readers as many MTBs do show 60° formations (as we also show in Supplemental Fig. 9), but these formations are observed to smaller extent in the amount of fabrication steps presented here. The authors expect 60° formations to be more prevalent with increased annealing/sputtering steps, as the formation observed in Supplementary Fig. 9 occurred after an SAAstep followed by an overnight anneal at 250 °C. In fact, the 1D structures have been shown to be intermediate strained phases², where the 60° formations are relaxed with little to no strain. However, again, our measurements do not indicate the formation of vacancy lines, which is expected as we are annealing during all steps of the

fabrication process and, as such, providing sufficient energy for atomic migration in the system. Our imaging data would suggest some amount of strain imparted to the 1D ID defect. This, however, is difficult to quantify as our images are limited by size, number of pixels, and dwell time needed over each point for sufficient signal to noise. Additionally, any drift from slight thermal fluctuations or piezo hysteresis during data acquisition may play a role. As such, we are confident in the structure assignment (which is well-matched to the expected electronic structure), but quantifying the amount of strain in such systems would involve another complete study with extensive ncAFM combined with DFT simulations.

> The exact formation of 1D IDs under the synthetic procedure we show and their subsequent evolution into 60° MTBs is very interesting and we are currently preparing a follow-up manuscript on this topic, since it goes beyond the scope of this paper. The key message of this manuscript is that a 1D inversion symmetry found also found in MTBs is not sufficient to create a TLL. It is the heterostructure and direct contact with graphene that is necessary to create the local quantum phase transition into a correlated electron system (i.e., a TLL) that does not require ultra low temperatures. In following we highlight the changes in the manuscript that make the discussion on what exact 1D system we are studying and effects much clearer to the reader.

. In conclusion, some of the presented results might correspond to MTB, whereas the images showing clear finite length certainly do not. This is what I stated in my previous report and since the authors have not taken this into account in the revised manuscript, it still cannot be considered for publication in Nature Communications.

Version 3:

Reviewer comments:

Reviewer #2

(Remarks to the Author)

The revised version of the manuscript contains no major changes.

The authors claim that the isolated 1D defects are mirror twin boundaries (MTB) or something the authors called "inversion-like domain" (ID). As I wrote in my previous report, this is obviously not possible. I asked the authors to provide a model for their 1D ID/MTB, but they did not. I hoped that had they attempted they would have realized it is not possible.

The authors try to support their claims by citing literature in rebuttal letter (and in manuscript):

"The reviewer again claims that finite length mirror twin boundaries are not possible. However, both Lehtinen et al. and Lin et al. have shown that 1D vacancy lines form into inversion-like domains which have been shown to be highly strained in the isolated 1D form^{1,2}."

There is no "isolated 1D form" in either Lehtinen et al or Lin et al. In both works, the finite length defect is always a vacancy line. The MTBs generated from the vacancy lines are always triangular loops. If not generated from vacancies (chalcogen deficiency), the MTBs cross the whole sample as grain boundaries.

Normal or shear strain does not help in achieving lattice inversion. Disclination-induced 60 degree bend of the lattice should be easy to confirm in the STM/ncAFM images, but there is no indications of this (and has never been reported previously).

"Our data would agree with these works and the multitude of others, where 1D chalcogen depletion eventually forms into mirror twin boundaries either across the entire flake or in triangular formations."

And all those works show that the intermediate phase is vacancy line. If the results are consistent with previous works, then the defects should be vacancy lines.

In the end, I want to note that vacancy lines and mirror twin boundaries have very different structure and properties and they cannot be mixed or grouped into a general "1D defect".

There is not more I can say. I think the manuscript is not suitable for publication Nature Communications.

Reviewer #4

(Remarks to the Author)

I have gone through the manuscript and the rebuttal letter for 2 rounds of review. It looks like the main debate is the atomic structure of the finite line observed by STM. Authors are arguing that they are symmetric inversion domain (SID), while they also observed some triangular shape of defects and they called that asymmetric inversion domain (AID). The original reviewer don't believe that the SID is MTB and consider it is just a line of vacancy complex.

Here is my opinion. From structural point of view, I totally agree with the reviewer. It is hard to imagine that a finite 1D line defect would be MTB, though the authors argue that it can be accommodated from defects at the end of the line. I have never seen such kind of structure and it would cause greater strain between the two sides since the orientation need to revert back 60 degree not only at the end of the line defect but also across the entire flake. So I agreed that the finite 1D line defects should not MTB, otherwise the authors should give clear atomic structure.

However, I pretty appreciate the high quality data of the STM/STS and the nAPRES, and the conclusion is also interesting that worthy for publication. So I highly suggest the author to perform additional STEM/TEM experiments on the same sample to rule out the confusion of the atomic structures of the finite 1D defect. The worst scenario is that the author would find these finite 1D lines are not MTB but vacancy line, but it did not hurt the novelty of the work since no one have proposed that the 1D vacancy line can also host the TLL state. It make sense that it have been reported that the 1D vacancy line would also become metallic like the MTB. It also make sense that after annealing the existing S vacancy would reconstruct into either line defects or form inversion domain, so thiese two defects can co-exist. It is highly possible that the authors are observing the mixture of these two structures since they also see difference in the STS between the SID and AID. Anyway I found the suggestion by the original reviewer is important and need to be clarified by additional experiments to make the atomic structure clear.

Version 4:

Reviewer comments:

Reviewer #2

(Remarks to the Author)

The authors now agree that some of the 1D structure could be vacancy lines. These are generally called 1D metals (1DM) in the manuscript. I think that is fine, although less satisfactory than actually knowing the defect type.

About the MTBness of the 1DMs, in the response the authors write:

"Lin et al. does show an isolated 4|4E GB-like intermediate as shown above in Lin et al. Figure 2."

This seems to be one of the main reasons for the confusion. They indeed seem to call a double vacancy line (as clearly depicted in the atomic structure inset) "4|4E GB-like structure" due to its similar chain of 4-rings in STEM images. However, it is obviously not a GB or MTB since the lattice orientation is the same around it. These structures only become GBs/MTBs in Figure 3(c) of that paper. In my opinion it is wrong to call the isolated versions of these MTB-like (since they are not mirror twins and not really even boundaries).

This is still reflected in the following parts of the manuscript.

1. In the text: "If isolated MTB-like structures are formed, additional defects form (or combine) to help relax the lattice, such as Mo chains or chalcogen vacancies, at either the endpoints or within 1DMs, as demonstrated in Supplementary Fig. 5."

Again, there is no isolated MTB-like structures. You cannot have inverted lattice on the two sides of the 1DM just by adding some vacancies at the end points, because the inverted lattices cannot be smoothly connected around the end points. This requires connecting more MTBs to the end points, which makes them not isolated.

2. Also in the text: "Our analysis reveals that among the types of 1DM defects measured, the isolated MTB-like structures resemble intermediate defects in the process of forming fully relaxed MTB structures from VS defects 28,31,36."

Yes, they form from VS defects. No, there are not isolated MTB-like structures. Isolated vacancy-line structures are the intermediates as discussed in Ref. 31.

3. Supp. Figure 5(a-c), this is not "4|4-P like". (a) It seems like an isolated segment, and thus it cannot have inverted lattice on the two sides. (b) The images clearly show it is asymmetric perpendicular to the 1DM, while 4|4-P would be symmetric.

4. Supp. Fig 3, these are all primarily vacancy lines as they are isolated segments. Also, in the caption "(b) 3 two SAA_step" should likely be "(b) three SAA_step".

5. Supp. Fig 4. These might indeed be as assigned, although a bit difficult to tell from the small figures.

Other comments not related to MTB notation:

6. In the text: "These defects are characterized by their distinct electronic behaviors—most are tungsten rich and exhibit electron-like characteristics, while sulfur rich 1DMs demonstrate hole-like behavior (see Supplementary Fig. 6)."

One cannot directly deduce the electron- or hole-like behavior from the stoichiometry (S-/W-rich). In principle, vacancy lines, 4|4P and 4|4E MTBs are all W-rich. It is also not clear how the S/W-richness is deduced here as Supp. Fig. 6 only shows the electron-/hole-likeness.

7. "Dense STS is acquired over an anticipated electron-like 1DM in Supplementary Fig. 16, where there is consistently no Egap opening near the EF across the entirety of the defect."

There are no visible 1DM states in the few-layer WS₂ spectrum so how can you claim there is no Egap opening?

8. Fig. 1(d), for the sake of completeness, it would be nice to see STS LDOS spectrum from a single vacancy on the multi-layer WS₂, if the authors happen to have one measured.

In conclusion, I think I now understand where the confusion related to the MTB-like structures arises from, but in my opinion it is still wrong to call them MTB-like. Since "...the primary focus of this manuscript lies in demonstrating the role of the graphene substrate in enabling correlated electronic states in WS₂", it is probably fine to call these defects simply 1DMs.

However, it is then unfortunate that the evidence for TLL/correlated states is not very strong: spin-charge separation in Figure 2 is far from clear and the lack of any signal from the 1DM on few-layer WS₂ makes it difficult to conclude much. For these reasons I would generally think that the manuscript does not meet the criteria for publication in Nature Communications, but at least I do not see major problems in it any more (after the authors have fixed the remaining issues mentioned above).

Reviewer #4

(Remarks to the Author)

I have gone through the rebuttal letter and the revised manuscript. I found the authors have addressed my comments and concerns nicely. The new data broadened the conclusion of TTL transition in 1D metallic defects including various structures like MTB, vacancy lines, etc., all of which presumably come from the underlying graphene substrate. I would like to recommend its publication. A minor point is that I suggest the author integrate the Fig. S12 in SI into the main text. This is very conclusive data showing the broadened claim of the revised manuscript.

Reviewer #1:

I read the paper with great interest, the authors investigated the WS₂'s MTB formed on graphene, which shows TLL properties. The form of TLL nature is related to the charge transfer between MTB and the Graphene. The work has its significance in showing us the influence from the substrate in formation of TLL, however, it's not clear (at least to me) that the gap renormalization is due to 1D correlation in MTB or due to MTB's other influence. Other comments is listed as follows

Thank you for your thoughtfulness and constructive comments. The reviewer has understood the main point we try to make in the manuscript, which is the importance of the environment (interface with graphene) that would lead to create a TLL in MTBs. Many of the reviewers following comments are inline with clarifying this point, which we have addressed. In addition, we modified the description of comparing STM and nARPES data to make the influence of graphene and charge transfer of graphene into the MTBs more clear. Also, we performed experiments on MTBs that sit on bilayer WS₂ with no direct contact to the graphene, where we do not observe the formation of a TLL (as shown in the newly revised manuscript).

1. Fig.1 show the as-grown WS₂ and the one sputtered by Ar ions. I guess the authors want to show us the comparison between the two samples, but the figures are shown with different scale bar, making the comparison less intuitive. Besides, the Fig.1b is not enough clear, although I could identify the 1D TLL and the vacancy, I still suggests the authors could add some marks on the figure.

We very much appreciate your input. Fig. 1 has been updated with same-size images across as-grown WS₂ and defective WS₂, where the image in Fig. 1 (a) has been updated. Additionally, we have marked where there are MTBs and V_S within Fig. 1 (b).

Fig. 1: WS₂ Defect Introduction. (a) Scanning tunneling micrograph depicting pristine WS₂ before Ar⁺ bombardment ($I_{\text{tunnel}} = 30$ pA, $V_{\text{sample}} = 1.2$ V). (b) After the pristine sample is heated and exposed to an Ar⁺ sputter ($I_{\text{tunnel}} = 30$ pA, $V_{\text{sample}} = 1.2$ V), both V_S and MTBs are present. Scale bars, 2 nm. (c) Lattice structure is measured by nAFM ($V_{\text{sample}} = 0.0$ V), with a CO-functionalized tip, which showcases the SMTB that is placed next to a structural schematic (d) of the MTB. Scale bar, 0.25 nm. (e) Point spectroscopy, in the form of the LDOS, for unmodified WS₂, V_S, and an SMTB. States around E_F for the SMTB are labeled as HOS-1, HOS, and LUS.

2. The spin-charge separation in E-K relation revealed by STM is not easy to identify in SFig.8 (spin's E-K), especially compared to Feng Wang's results (Nat Mater, 21, 748).

Thank you for making this comparison, which was also raised as a cause of concern from additional reviewers. We do apologize for making this less obvious in the first submission draft (previously

Supplementary Fig. 7). We have increased the energy range measured and removed all filtering from the as-measured STS dispersion (outside of normalizing the y-scale of each STS curve). Charge and spin excitations are now clearly visible above the LUS, which exhibit different spatial frequencies that are highlighted in the FT-STs. We also extract line profiles across a given energy to visualize behavior at the HOS, LUS, LUS(s)+1 and LUS(c)+1, which are most easily distinguishable in recent data. Here, the calculated mode of peak-peak distances of the first spin excitation and first charge excitation show a measurable difference and help us highlight these two peaks in the FT. Additionally, the branching region is better distinguishable in the FT above the LUS beyond the first excitation. One extra comment is that we work in close collaboration with both Mike Crommie's group and Feng Wang's groups on many related projects, and we have found that the creation of longer MTBs in WS₂ is more challenging than MoSe₂ and MoS₂, the systems which have been used to study TLLs in the groups of Mike Crommie, Feng Wang, Achim Rosch, and Thomas Michely. In the shorter MTBs, less excitation nodes make the signal-to-noise ratio in the FFT less, as we are frequency limited, than the longer MTBs available in MoSe₂ or MoS₂. The reason we were focused on WS₂ is two-fold, 1) show that the tungsten-based TMDs, using the newly introduced Ar⁺-ion fabrication techniques, can create 1D systems that exhibit TLL behavior and 2) WS₂ specifically has a very strong spin-orbit coupling amongst available TMD systems, which makes TLLs formed in WS₂ a very interesting system to be coupled to other correlated electron systems for study of related phenomena.

Fig. 2: Dense LDOS Mapping. (a) An SMTB is measured topographically, where the red line shown depicts a length of 11.19 nm ($I_{tunnel} = 30$ pA, $V_{sample} = 1.2$ V). Dense local density of states spectra is collected along (a) an SMTB (1x128x500 pixels) ($V_{modulation} = 5$ mV, $I_{set} = 150$ pA) beginning at the starred point along the red line. This is shown as a (b) function of bias and distance. The LDOS dispersion displays spin-charge separation that is identified in the first spin and charge excitations (c) at 0.060 V (blue) and 0.072 V (red). A 1D particle-in-a-box behavior is also present at -0.068 V (orange), 0.036 V (green), and 0.060 V (blue), which are highlighted in (b). The number of nodes (N) increases linearly (N+1) and is shown (c) from the HOS to LUS+1 (maxima are starred). Charge and spin levels at LUS+1 exhibit the same number of nodes, however the most frequent peak-to-peak distance is 1.85 nm and 1.59 nm, respectively. An FFT of (b) is shown in (d), where the LUS(s)+1 and LUS(c)+1 show a separation onset, and both the spin (blue) and charge (red) branches can be monitored up to 0.5 V (maxima labeled as starred points). A static branch is also measured at 0.48 nm^{-1} .

3. The correlation gap near E_f in MTB is highly unsymmetrical about E_f , E_f is almost at the edge of HOS, why

Excellent question and thank you for asking. The HOS-LUS energy gap can fluctuate around E_F as doping level is modified¹. Here, the gap can rigidly shift around E_F in both hole-doped (gate voltage < -10 V) and also in the electron-doped regime (gate voltage > 10 V). For our system, there exists a substrate-induced pinning of the WS_2 E_F to the upper third of the band gap near the CBM². Tip contributions and neighboring defects can also induce energy level effects as mentioned in reference to the updated Supplementary Fig. 5. We have also added a line into the main text addressing this question.

The HOS-LUS position around the E_F can shift and is driven by, e.g., neighboring defects, tip-induced effects, and substrate variation^{1,2}.

4. The shift of E_f in Graphene is attributed to the transfer charge to MTB, but not charge transferred from graphene to the V_s . Is this because the unoccupied states for V_s locate at much higher level?

Very good point and we appreciate the opportunity for further clarification. As V_S in-gap states are unoccupied and located above the E_F , there is no valence band overlap for electrons to transfer into unoccupied states without additional excitation. We note that MTBs show similar band renormalization as measured via nARPES. Overall, V_S defects measured do not exhibit such behavior. Lastly, the sample was annealed in the nARPES chamber for 12 h at 250 C after transfer from an Ar (low-oxygen environment) chamber, which would play an added role into the amount of V_S present within the sample. The below statement was added to the main text and the methods section has been updated.

We can exclude a charge transfer involving the single V_S , which has its characteristic in-gap state above Fermi level and therefore unoccupied, as displayed in the STS curve in Fig. 1 (e).

5. Can the correlated gap observed by STM be fitted with power-law function in TLL model.

Thank you for your question. Suppression of the DOS was shown to follow a power-law behavior for defects within the range of > 30 nm in Xia et al.³, however, the procedure presented creates defects within the range of 8 - 14 nm. The authors do postulate that MTBs of increased length can be grown with increased SA/SAA steps. As such, our suppression near the HOS-LUS energy gap is more consistent with Zhu et al.⁴ and Jolie et al.¹ due to nature of defects produced. Within the length regime measured, there is additional suppression of the DOS due to finite defect length and other contributions driven by neighboring defects or other bounded impurities that is predicted by Luttinger liquid theory and described by Meden et al⁵. We add this sentence to the text to better address your comment.

We also look at the DOS measured in topographic STM images, as created defects are, on average, below the size regime shown to yield power-law scaling in point LDOS spectra^{3,5}.

6. some typos, like in first line in discussion part

Thank you for noticing, as this was oversight on our part. The first sentence of the discussion has been modified accordingly. Other typos that were noticed have also been corrected.

This work presents a new and controllable way to create MTBs in WS_2 epitaxially grown on graphene.

7. Discussion part should include more discussions about mechanism that how transfer charge between Graphene and MTB induce the gap renormalization.

The gap renormalization arises as the self-energies of the band-edge states shift due to Coulomb interactions among free carriers⁶. In our case the presence of MTB that displays a metallic behavior screen the coulomb interaction of the WS_2 carriers leading to a smaller TMD gap. The following sentences have been added to the main text and discussion.

The gap renormalization arises as the self-energies of the band-edge states shift due to the Coulomb interaction among free carriers⁶. The presence of metallic MTBs screen the coulomb interaction in the WS_2 crystal leading to a smaller TMD gap.

We observed how defective states behave as an acceptor of graphene electrons and that they cause a massive band gap renormalization of WS_2 , where the presence of MTBs screen the coulomb interaction of WS_2 carriers leading to an overall smaller TMD gap.

8. Some other works relating to TMD's edge state should be referred Proc. Nat. Acad. Sci. USA 113, 8583–8588 (2016) Nat. Commun 11, 659 (2020)

We very much appreciated your suggestions in adding these references related to edge-state behavior within TMDs. Both references have been added.

One-dimensional (1D) systems in condensed matter physics provide unique insight into a variety of quasi particle excitations, including charge density waves (CDWs) that arise due to Peierls instabilities^{7,8}, lossless transport through electronic wires in topological edge states^{9–11}, quantum spin liquids¹², as well as more exotic phenomena such as Majorana modes in nanowires¹³ and the emergence of a Tomonaga-Luttinger liquid (TLL).

Reviewer #2:

The authors study electronic structure of mirror twin boundaries (MTB) and Tomonaga-Luttinger liquid (TLL) behavior in particular, in WS₂ using scanning probe microscopy and nARPES. MTBs are introduced to WS₂ by Ar⁺-ion bombardment and annealing. The authors claim to elucidate on how the substrate affects TLL and find charge transfer from graphene to MTB, which in turn leads to band shifts.

In my opinion, the insights gained onto electronic structure and TLL of MTBs in this manuscript are not new and the presented results are not of particularly high quality (compared to those already reported in the literature). Ar-ion bombardment induced MTB formation and nARPES data of such samples might be new, but not enough to warrant publication in Nature Communications. In more detail:

We thank the reviewer a lot for the detailed comments because we feel it enhanced the clarity of the manuscript, as detailed below. We also recognize that our main new findings were not portrayed sufficiently clear. While Ar⁺-ion bombardment to induce MTB formation and the nARPES are novelties for this system, we agree with the referee that these are perhaps more technical. Our main new scientific insight is the importance of graphene as a direct contact to these MTBs that help form a TLL in the first place, where a quantifiable shift of the graphene Dirac point compared to the Fermi level is visualized. We have clarified the discussion of the correlation data between STM, STS, ncAFM, and nARPES to make this point more clear and, in addition, we performed new experiments on MTBs in contact with WS₂ bilayers on the exact same system, showing there is no formation of a TLL. After describing our main scientific insight more clearly in the manuscript and supporting the claim with additional experiments, the reviewer also had specific comments that helped both the general clarity of the original manuscript and underlying full scientific finding. We understand that the quality of the data is not as clear, as we work in close collaboration with both Mike Crommie's group and Feng Wang's groups on many related projects, and we have found that the creation of longer MTBs in WS₂ is more challenging than MoSe₂ and MoS₂, the systems which have been used to study TLLs in the groups of Mike Crommie, Feng Wang, Achim Rosch, and Thomas Michely. In the shorter MTBs, less excitation nodes exist, which make the signal-to-noise ratio in the FFT less, as we are frequency limited, compared to the longer MTBs available in MoSe₂ or MoS₂. The reason we were focused on WS₂ is two-fold, 1) show that the tungsten-based TMDs, using the newly introduced Ar⁺-ion fabrication techniques, can create 1D systems that exhibit TLL behavior and 2) WS₂ specifically has very strong spin-orbit coupling amongst available TMDs, which makes TLLs formed in WS₂ a very interesting system to be coupled to other correlated electron systems for the study related phenomena.

1. TLL is not conclusively demonstrated.

Thank you for your comment, as further investigation was also suggested by other reviewers. We were able to revisit our previous data and also collect a new set of data of experiments performed after reviews were received. Latest analysis is updated in Fig. 2 (previously Supplementary Fig. 7), where all filtering

of the STS is removed (outside of normalizing the y-scale). We also help clarify what criteria define a TLL in Supplementary Note 4, and hope the sum of these efforts helps address your concern.

Fig. 2: Dense LDOS Mapping. (a) An SMTB is measured topographically, where the red line shown depicts a length of 11.19 nm ($I_{tunnel} = 30$ pA, $V_{sample} = 1.2$ V). Dense local density of states spectra is collected along (a) an SMTB (1x128x500 pixels) ($V_{modulation} = 5$ mV, $I_{set} = 150$ pA) beginning at the starred point along the red line. This is shown as a (b) function of bias and distance. The LDOS dispersion displays spin-charge separation that is identified in the first spin and charge excitations (c) at 0.060 V (blue) and 0.072 V (red). A 1D particle-in-a-box behavior is also present at -0.068 V (orange), 0.036 V (green), and 0.060 V (blue), which are highlighted in (b). The number of nodes (N) increases linearly (N+1) and is shown (c) from the HOS to LUS+1 (maxima are starred). Charge and spin levels at LUS+1 exhibit the same number of nodes, however the most frequent peak-to-peak distance is 1.85 nm and 1.59 nm, respectively. An FFT of (b) is shown in (d), where the LUS(s)+1 and LUS(c)+1 show a separation onset, and both the spin (blue) and charge (red) branches can be monitored up to 0.5 V (maxima labeled as starred points). A static branch is also measured at 0.48 nm^{-1} .

Supplementary Note 4

A TLL low-energy Hamiltonian in a box can be defined as^{1,4,14-17}

$$H_{TLL} = \frac{\pi v_c N^2}{4LK_c} + \frac{\pi v_s S_s^2}{LK_s} + \sum_{n=1}^{\infty} (v_c k_n a_{c,n}^\dagger a_{c,n} + v_s k_n a_{s,n}^\dagger a_{s,n}),$$

where L is defect length, c and s label the charge and spin channels, K_c and K_s are two Luttinger parameters, v_c and v_s are charge and spin velocities, N is the total electron filling, S_z is the total z -component spin number, and $a_{c,n}^\dagger$ and $a_{s,n}^\dagger$ are the creation operators of charge and spin excitation. This representation identifies the key requirements for a TLL to be present within a MTB. These requirements are such that a) the first term defines a charging energy that determines the HOS-LUS energy gap (E_{gap}) arising from Coulomb interactions, b) the second term defines the spin sector where the ground state has zero spin, c) spin and charge show independent dispersions, and d) there exists E_{gap} dependence on MTB length. In the results presented, all requirements are fulfilled to showcase the presence of a TLL hosted by a MTB within WS_2 .

The change in the periodicity isn't particularly evident in Figure 2 or SI Figs 5 and 6. "We also highlight nodal increase as a function of bias in Fig. 2 (c), where the number of nodes moves from 6 at the LUS up to 10 nodes at 0.4 eV before nearing the CBM of WS 2 for the SMTB." Here, counting of nodes should refer to the nodes in the quantum well states, and the size of each node should remain about the same. In Figure 2(c), the size of the nodes seem to vary (if counted to have e.g. 6 nodes at LUS), apparently due to overlap of the atomic features and the quantum well states. It is then not clear how to find the number of QW-state nodes. In Figure 2(f), the sizes are consistent, but then the change in the number of nodes is only one: "This is also shown for an AMGB in Fig. 2(f), where 8 nodes are measured at the HOS and 9 nodes are measured at -0.3 eV."

Thank you for your comment. We do apologize for not being very clear in our dI/dV measurements (which we have also corrected the excitation levels in prior Fig. 2, now Supplementary Fig. 5). Updated Fig. 2, shown above, now better showcases the number of quantum-well state nodes above and below the TLL energy gap. The MTB in Fig. 2 is better isolated and unperturbed, and thusly behaves as expected (showing integer behavior). Supplementary Fig. 5 now contains better clarification of the particle-in-a-box behavior being presented with added extracted line profiles (as shown further below) of the regions depicted in Supplementary Figs. 6 and 7 (earlier Supplementary Figs. 5 and 6). Lastly, we note that one of the defects shown in Supplementary Figs 5 and 6 is neighboring a sulfur vacancy, which splits the number of QW-state nodes in as-acquired data above the HOS, as the number of nodes from the HOS to the LUS, in this unique case, follows $n+2$ behavior (where the QW-state central node relative to the defect length is scattered by a neighboring sulfur vacancy).

Supplementary Fig. 5: **LDOS Mapping**. Conductance maps ($V_{modulation} = 5$ mV) performed across an SMTB show dual-line orbital behavior at (a) 0.132 eV (LUS) and (b) -0.025 eV (HOS) that is spatially out of phase. (c) Accumulated conductance maps across a single-line of the SMTB ($V_{modulation} = 5$ mV) are further shown as a function of bias, where the number of nodes increase as bias voltage is increased. Scale bars, 1 nm. Peak assignments can be made by solving for local maxima along a line profile, which is compiled in (d). Conductance maps (dI/dV) of the as-measured (e) 0.047 eV (LUS) and (f) -0.047 eV (HOS) that are spatially in phase within an AMGB. (g) Compiled conductance maps ($V_{modulation} = 5$ mV) across the single-line AMGB are further shown, where the number of nodes decreases as the voltage is increased. (h) Line profiles extracted from (g) detailing local maxima.

Supplementary Fig. 6: **Differential Conductance Mapping.** dI/dV mapping ($V_{modulation} = 5$ mV) over the spectra region shown for an SMTB on a jet color scale, where the energy is ramped from near the VBM of WS_2 by 0.05 V to the HOS gap opening of the MTB hosting a TLL, and then from the LUS to the CBM of WS_2 . Arrows indicate decreasing bias. The defect imaged is shown to the upper left, where dI/dV images are representative of the region highlighted in dashed red ($I_{tunnel} = 30$ pA, $V_{sample} = 1.2$ V). Scale bar, 1.5 nm. A 1D particle in a box behavior is evident, and orbitals of both the TLL and a V_S can be visualized on as-acquired data at respective energies. Additionally, presence of a V_S scatters available quantum-well states (spatially-centered within the defect) above the HOS.

Supplementary Fig. 7: **Differential Conductance Mapping.** dI/dV mapping ($V_{\text{modulation}} = 5$ mV) over the spectra region shown for an AMGB on a jet color scale, where the energy is ramped from near the VBM of WS_2 by 0.05 V to the HOS, and then from the LUS to the CBM of WS_2 . The defect imaged is shown to the upper left, where the partial portion of the mapped defect is highlighted in dashed red ($I_{\text{tunnel}} = 30$ pA, $V_{\text{sample}} = 1.2$ V). Scale bar, 4 nm. Orbitals of the formed TLL can be visualized on as-acquired data as a function of bias voltage, where arrows indicate decreasing bias.

Supplementary Figs. 6 and 7 further show conductance maps as a function of energy both below HOS and above the LUS for both structures, where the presence of a neighboring V_S scatters quantum-well states above the LUS and the number of observable nodes from the HOS to the LUS increases from 4 to 6 nodes, respectively, and this then increases from 6 at the LUS up to 10 nodes at 0.4 eV before nearing the CBM of WS_2 .

In SI Fig 5 it is difficult to see how many nodes there are at each bias voltage. In SI Fig 6 the number of nodes changed from 6 at $V=600$ mV to 8 at $V=-250$ mV, and it is difficult to read outside this range. The overall picture is easier to grasp from Figure S7, but the quality is mediocre. The spin- and charge-band dispersions in Fig. S7(c) are a leap of faith, as is the DOS fit in Figure S9.

Thank you for your comments that are very appreciated, which we believe have overall helped us make the presentation more clear. The spin- and charge-band dispersions are more obvious now in the first excitation above the LUS in Fig. 2 (b and c), shown above, with all 1D spectra filtering removed and the energy range of visualization increased. Also, the FT-STIS more clearly shows a branching region (spin-charge separation) as labeled. SI Fig 5, also shown above, has been updated to include line profiles to better highlight the number of nodes at as-acquired energy levels. We made additional experiments to include measurements of MTBs of over multilayer WS_2 . We find spectroscopically no evidence of formation of a TLL shown in a new main text Fig. 5 and Supplementary Fig. 12. We also followed the argumentation of Xia et al. to use line profiles of the DOS at the onset of the MTB³, where you can extract the Kc value. We have done that for the single-layer MTB (within monolayer WS_2 /Graphene/SiC(0001)) and the multi-layer MTB (within multilayer WS_2 /Graphene/SiC(0001)) in the latest Supplementary Fig. 9. We have found a Kc of 0.5 in the monolayer case, which is what we expect, and have found the Kc value of near 1 for the multi-layer (where the MTB is in contact with underlying wide band gap material).

Fig. 5: **LDOS on Multilayered WS_2** . (a) Scanning tunneling micrograph depicting 3 ML of WS_2 over MLG/SiC(0001) with both MTBs and point defects ($I_{tunnel} = 30$ pA, $V_{sample} = 1.2$ V). Scale bar, 4 nm. (b) dI/dV point spectroscopy of 3 ML WS_2 (black), an SMTB within 1 ML WS_2 (red), and an SMTB within 3 ML WS_2 (blue). Corresponding spectral locations are depicted in (a), where both the VBM onset and CBM onset for the SMTB (blue) match that of the as-grown 3 ML WS_2 (black) spectra.

Supplementary Fig. 12: **Spatially Resolved Scanning Tunneling Spectroscopy over Multilayer WS₂.** Dense LDOS spectra (1x128x500 pixels) collected over an SMTB ($V_{modulation} = 5$ mV, $I_{set} = 150$ pA), where states within the gap are not present.

Supplementary Fig. 9: **Constant-Current Density of States Decay Across an MTB.** A power-law dependence is measured across a multiple MTB defects on both monolayer WS_2 and multilayer (3ML) WS_2 on $\text{MLG}/\text{SiC}(0001)$. In each image depicted, multiple fits were taken across defects shown, where representative linescans beginning at a starred point and along a given line (red) are shown with corresponding constant-current profiles with subsequent exponential fittings. Across 14 lineprofiles and 10 defects, multilayer WS_2 defects show an absolute power-law exponential parameter of 0.99 ± 0.15 , which indicates Fermionic behavior, and monolayer WS_2 defects yield a parameter of 0.49 ± 0.16 that matches expected behavior of a Luttinger liquid. Fittings were performed using the lmfit package in Python¹⁸.

The presentation is lacking in places. Figure 2, it is not clear if the whole MTB is shown or just a small part of it? In (c) and (d), it is not clear what is shown, is it a collection of conductance maps or is x-axis position coordinate along the MTB and y-axis bias voltage?

We agree and apologize that Fig. 2 was not very clear. The regions measured are the same as presented in Supplementary Figs. 6 and 7, which now also contain topographic images depicting where LDOS measurements were taken within the image (where updated figures are shown in earlier discussions). Prior Fig. 2 (c and f), now Supplementary Fig. 5 (c and g), show cropped dI/dV image regions to highlight nodal behavior at a given energy.

2. All of those results related to TLL have already been presented in the literature, but in higher quality with all these features clearly resolved, e.g., in Ref. 16 where also the graphene doping is studied via gating.

We certainly agree that the literature has shown TLL behavior within MTBs hosted by MoS₂, MoSe₂, and WTe₂, however, this phenomena in WS₂ has yet to be examined due to the formation energy of sulfur vacancies within WS₂. We hope you will reconsider given the latest round of revisions and additional experiments performed. Again, we do very much appreciate your thoughtful comments which have helped us reorganize the presentation of our data and make the paper overall better. We studied the influence of doping using graphene as one of the electrodes in Zhu et al.¹, however, we were not aware of the impact of graphene in itself as the interface that is critical to form the TLL, which is what we found here and crucial for this finding was the ability to do correlative measurements between STM and nARPES, as well as comparing the LDOS of the exact same system but sitting on a bilayer. The graphene has looked into what is the effect of the doping when you put more electrons into the system, but it was not very clear that the graphene itself was critical. nARPES data show a measurable Dirac point shift, enabling the visualization of the direct charge contribution to the MTB from graphene. The quality of the FFT data was less because we are frequency limited, as mentioned in more detail above. Also, we wanted to see that MTBs in WS₂ form a TLL. WS₂ itself has strong spin-orbit coupling. Length is also a limiting factor, therefore the resulting signal-to-noise ratio for the FFT is limited. We have better addressed this point by updating our analysis in, now, Fig. 2, where spin and charge separation is shown with greater detail.

The charge transfer from graphene to the 2D materials is discussed (in more detail) in, e.g., C. Murray et al., ACS Nano 14, 9176 (2020). In fact, the authors do not seem to discuss the band banding issue here at all, but simply assume that the charge transfer from graphene to defective WS₂ yields a homogeneous band shift, which I would think is incorrect.

We appreciate the opportunity to clarify this point and for your suggested reference, which has been added to main text. First, we have presented localized band bending in Fig. 4. STS across the MTB show how the VBM and CBM do not produce a homogenous shift, but rather gradually shift as the STM tip approaches the MTB. We have also added the below comment to the text. Furthermore, we quantify the charge transfer using nARPES to identify the shift of the graphene Dirac point compared to the Fermi level. In the correlative study on the exact same sample, we also clearly show via STM that the charge transfer from the graphene goes into the MTBs and not into any of the other intrinsic defects that are present. See our description and discussion of Fig. 4.

Fig. 4: **Tomonaga Luttinger Liquid Spatial Dependence.** (a) LDOS spectra collected from pristine WS₂ to an MTB ($V_{modulation} = 5$ mV, $I_{set} = 150$ pA) and then recorded in reverse. The as-measured VBM, CBM, HOS, and LUS are highlighted with relative positions to E_F . (b) Overview of the process of charge transfer from graphene into an MTB that is able to host highly correlated electron states is depicted, where the E_F of graphene is shifted and locally modified due to the presence of an MTB.

Spatially-resolved STS spectra reveal a band bending across the MTB, akin to results by Murray et al.¹⁹ and the pristine WS₂ that extends over 500 pm, giving rise to a sharp junction across the two regions.

3. The authors don't obtain insight on the role of substrate on the electronic structures. The authors only compare samples without MTBs to samples with MTBs on the same substrate and thus it is difficult to draw conclusions on how the substrate affects MTBs. For that there should be a comparison between different substrates.

Thank you very much for your comment regarding different substrates. Other groups have studied MTBs formed in TMDs on a variety of different substrates, which we have also been cited in the manuscript. Hence, we decided to look at the same system with MTBs that are formed on bilayers using the same Ar⁺-ion bombardment procedure. We clearly see no formation of TLL within these 1D defects. In order to better determine the role of the substrate outside of comparing pristine WS₂ to WS₂ with a number of MTBs, we decided to collect another round of data comparing as-formed MTBs on multilayered WS₂. The result showcases Fermi liquid behavior above the CBM rather than a TLL. Additionally, there are no

states within the gap, as there is no direct contact with underlying graphene. This is now highlighted in Fig. 5 and Supplementary Fig. 12 (repeated below).

Fig. 5: **LDOS on Multilayered WS₂**. (a) Scanning tunneling micrograph depicting 3 ML of WS₂ over MLG/SiC(0001) with both MTBs and point defects ($I_{tunnel} = 30$ pA, $V_{sample} = 1.2$ V). Scale bar, 4 nm. (b) dI/dV point spectroscopy of 3 ML WS₂ (black), an SMTB within 1 ML WS₂ (red), and an SMTB within 3 ML WS₂ (blue). Corresponding spectral locations are depicted in (a), where both the VBM onset and CBM onset for the SMTB (blue) match that of the as-grown 3 ML WS₂ (black) spectra.

Supplementary Fig. 12: **Spatially Resolved Scanning Tunneling Spectroscopy over Multilayer WS₂**. Dense LDOS spectra (1x128x500 pixels) collected over an SMTB ($V_{modulation} = 5$ mV, $I_{set} = 150$ pA), where states within the gap are not present.

Moreover, there is no data on how the TLL is affected by doping and while the defective samples contain both vacancies and MTBs, their relative roles to the doping (and TLL) cannot be separated. As a minor

note, the authors write that MTBs with increased length are formed in SAA_{step} (as compared to SA_{step}), but why the Table in Suppl. Fig. 1 shows decreased defect density for SAA_{step} .

Where our sample is not gated, we are able to determine that monolayer TMD on graphene plays a key role, and there is no in-gap opening from multilayer TMD. We certainly agree that fabricating a gated device with the presented system could add subsequent information of the TLL formed in WS_2 (with an additional comparison to multilayered WS_2), but this would accumulate into another report to best address the behavior of a gated system. The main point we try to address here is the importance of graphene/MTB in WS_2 for the formation of a TLL (where a clear shift of the graphene Dirac point compared to the Fermi level is present). In regards to defect density in the table in Supplementary Fig. 1, we present the local picture measured with LTSTM. We have added the statement below in the main text for clarification. Lastly, the sample was annealed in the nARPES chamber for 12 h at 250 C after transfer from an Ar (low-oxygen environment) chamber, which would play an added role into the amount of V_S present within the sample measured in nARPES. We note that MTBs show similar band renormalization as measured via nARPES. Overall, V_S defects measured do not exhibit such behavior. The below statement was also added to the main text and the methods section has also been corrected to better reflect our findings. Something else to note, is that we are currently in the process of studying MTB formation in LEEM, however this goes beyond not only the manuscript here but also the capabilities of using STM in an efficient manner.

Densities of MTBs are also reduced, to a lesser extent compared to V_S , from the SA_{step} [0.012 to 0.030 MTB/nm²] to the SAA_{step} [0.008 to 0.010 MTB/nm²], as MTB length is increased via annealing

We can exclude a charge transfer involving the single V_S , which has its characteristic in-gap state above Fermi level and therefore unoccupied, as displayed in the STS curve in Fig. 1 (e).

Reviewer #3:

This manuscript reports an interesting method to create mirror twin boundaries (MTBs) in WS_2 monolayers. As I learned from the manuscript that the large formation energy for S deficient structures in WS_2 hinders the formation of MTBs in it. The authors used low-energy Ar+ sputtering to create S vacancies for substantially reducing the S concentration in WS_2 and then annealed the sample to form MTB defects, which promote the formation MTBs. Given the prepared MTB structures, the authors found that charge transfer from the graphene substrate, providing weak electronic screening, enhances electron-electron interactions in MTBs and thus forms TLL states within 1D MTBs. Although the preparation of MTBs in WS_2 seems novel to me, observations of TLL states in MTBs of other TMDs were previously reported (e.g. DOI: 10.1103/PhysRevX.9.011055, 10.1021/acsnano.0c05397), also from the group(s) of some of the authors. In light of this, I was not fully convinced that this manuscript is justified for publication in Nature Communications, especially in the absence of discussions on the growth process and mechanism. My detailed comments are as follows.

We very much appreciate your insight. The reviewer has understood the difficulties associated with inducing MTBs in WS_2 and has helped provide additional comments that have helped clarify our scientific findings. Our main new scientific insight is the importance of graphene as a direct contact to these MTBs that help form a TLL in the first place, where a quantifiable shift of the graphene Dirac point compared to the Fermi level is visualized. When we studied the influence of doping using graphene as one of the electrodes in Zhu et al.¹, we were not aware of the impact of graphene in itself as the interface that is critical to form the TLL. We have clarified the discussion of the correlation data between STM, STS, ncAFM, and nARPES to make this point more clear and, in addition, we performed new experiments on MTBs in contact with WS_2 bilayers on the exact same system, showing there is no formation of a TLL.

1. As the two-step preparation protocol is novel for building MTBs in WS_2 , I strongly urge the authors provide more information on the growth mechanism. I appreciate that that the authors carried simulations

on the creation of S vacancies, but it is unclear that the kinetic process and the formation energy of S vacancies transforming into MTBs.

Thank you for your comment. Much of the kinetics and subsequent mechanisms have been explained in the literature²⁰⁻²³. We also add a supplementary note indicating what has been reported in this regard. We are actually very interested in understanding the formation better, also inspired the papers of Lehtinen et al. and Gibertini et al.^{21,22}, however this goes beyond not only the manuscript here but also the capabilities of using STM in an efficient manner. To this end, we are currently in the process of studying MTB formation in LEEM. Our goal here is not only to understand the kinetics better, but the ability to use the presented Ar⁺-ion approach to form MTBs with arbitrary length.

Supplementary Note 3

Formation kinetics of inversion domains have been investigated for a number of MX₂ structures^{20,22,23} and both the SMTB and the AMGB structures formation energies were shown to be described by the equation,

$$E_{MTB} = E_{Total} - n_M \mu_{MX_2} - \Delta n \mu_x - c,$$

where E_{Total} is the total energy of the system with MTB, n_M is the number of metal atoms, μ_{MX_2} is the energy of pristine MX₂ per formula unit, Δn is the number of missing chalcogen atoms, μ_x is the energy of chalcogen atoms, and c is the energies of the two edges on both sides of the mirrored ribbon²³. A large number of missing chalcogen atoms, induced by Ar⁺ bombardment, enable the formation of subsequent MTBs.

2. In Fig.1(e), there is another peak sitting at the right side of the HOS. does it arises from a folded state of HOS or represents another band residing below HOS? Please the authors clarify.

We apologize for not mentioning what this side peak is specifically. We have updated Fig. 1 to better address this.

Fig. 1: WS₂ Defect Introduction. (a) Scanning tunneling micrograph depicting pristine WS₂ before Ar⁺ bombardment ($I_{\text{tunnel}} = 30$ pA, $V_{\text{sample}} = 1.2$ V). (b) After the pristine sample is heated and exposed to an Ar⁺ sputter ($I_{\text{tunnel}} = 30$ pA, $V_{\text{sample}} = 1.2$ V), both V_S and MTBs are present. Scale bars, 2 nm. (c) Lattice structure is measured by nAFM ($V_{\text{sample}} = 0.0$ V), with a CO-functionalized tip, which showcases the SMTB that is placed next to a structural schematic (d) of the MTB. Scale bar, 0.25 nm. (e) Point spectroscopy, in the form of the LDOS, for unmodified WS₂, V_S, and an SMTB. States around E_F for the SMTB are labeled as HOS-1, HOS, and LUS.

3. I was curious why both the HOS and LUS were observable in STS solely, but were invisible in ARPES measurements, even at 6 K. Could the authors explain?

ARPES measurements are averaged over a $1 \mu\text{m}^2$ area. Within this region it is possible to find MTBs with a given length distribution. The HOS-LUS gap is a function of the MTB length, and this would

result in a very broad feature in ARPES spectra by itself. In general, the presence of defects results in a higher background signal and broader bands. In order to have a clear signal of the HOS-LUS bands, one should have a very ordered system where defects form a structure such as the one reported by Ma et al.²⁴, however, both the material and the growth technique employed by Ma et al. are different to what was reported in our manuscript. We have added the following comment in the main text to address this.

The HOS/LUS states, observed in STS, are not visible via ARPES experiment. The disorder dictated by the presence of defects results in a higher background signal and broader WS₂ bands. In order to have a clear signal of the HOS-LUS states, very ordered MTBs with homogeneous length and orientation are necessary²⁴.

4. All minus signs in the E-E_F label of Fig. 3(a) are missing; Those two shadowed bars in Fig. 3g were not explained in the associated caption and/or the main text.

Thank you for noticing this, which was an oversight on our part. Fig. 3 is now updated accordingly.

Fig. 3: nARPES Band Structure Comparison. (a) Unmodified and (b) defective band structure of WS₂ on graphene. The inset displays the WS₂ (green) and the graphene (black) BZ. The spectra are collected along the Γ -K orientation. nARPES spectra of as-grown (c) and defective (d) crystals that are collected along the red line of the inset in panel (a), with respective MDCs at the E_F shown in (e) and (f). (g) EDCs overlapped with STS from unmodified and defective WS₂. Dark blue and dark red are the STS signals collected on pristine WS₂ and on an MTB, respectively. The light blue and light red lines are the EDCs collected at the K point of the BZ (dashed vertical lines in panels (a) and (b) respectively). Vertical semitransparent bands display the onset of the VB for pristine (blue) and defective (red), corresponding to EDC kinks clarified with black arrows.

5. The authors compared linecuts of ARPES maps with STS spectra in Fig. 3(g) and claimed that they are well consistent. With due respect, I was not convinced that they are “consistent” from any energy range plotted in Fig. 3(g). Please the authors explain, by comparing which characteristic(s), they are consistent. In addition, I noticed that the linecuts of the ARPES maps hit the energy axis above E_F . Could the authors explain this?

Thank you for your comment and the opportunity to help further clarify the point raised. The nARPES energy distribution curves (EDCs) are collected along the K point of WS_2 BZ. The shaded area in Fig. 3 (g) highlights the onset of the WS_2 VB. An arrow has been added to the figure in correspondence of a kink in the EDC. A kink in the EDC is commonly associated with the onset of an electronic state. The VB onset energy value extracted this way is consistent with the top of the valence band observed in the spectrum of Supplementary Fig. 10 collected with a different polarization. We know from our presented procedure that we predominately produce two types of defects that may contribute to states formed within the gap of WS_2 measured with STM/STS. Additionally, with STM/STS, we measure local band bending across a MTB shown in Fig. 4 (a). We rule out the contribution of V_S in nARPES data by measuring the VBM onset of single MTBs in STS directly compared to nARPES EDCs of defective regions. As a matter of facts, the gap renormalization is absent for V_S defects. Fig. 3 (g) further highlights how the EDC of the defective sample is inconsistent with the STS spectra collected from a single V_S , shown in Fig. 1 (e). However, when overlapped with the STS from a MTB defect we can see that the onset of the VB matches the ARPES EDC. We therefore ascribe the shift of the valence band to the presence of MTBs. The following sentence has been added to the main text. Additionally, another portion of the EDC analysis has been updated. In response to your second comment, in the nARPES EDCs, data points above the E_F are included. This inclusion is due to a deliberate small gap maintained between the E_F and the edge of the detector during nARPES spectra acquisition. Although the absence of photoelectrons above the E_F is expected, maintaining this gap facilitates a more accurate estimation of the distribution across the E_F .

We can exclude a charge transfer involving the single V_S , which has its characteristic in-gap state above E_F and therefore unoccupied, as displayed in the STS curve in Fig. 1 (e).

An analysis of the energy distribution curves (EDC) intersecting the VBM is presented in Fig. 3 (g), denoted by the vertical dashed line in Fig. 3 (a-b). The EDCs are compared with STS curves obtained from the unmodified crystal and a 1D MTB formed on TMD. The pristine structure (shown as blue lines) exhibits a satisfactory alignment at the expected onset of the VBM. The vertical semitransparent bars, displayed in blue and red, highlight the energy region associated with the VBM onset for both the pristine and defective systems. A distinct kink in the EDC indicates the onset of the VBM as extracted from the nARPES data, marked by the black arrow. The observed band shift is primarily attributed to the band gap renormalization caused by the presence of the MTB, in contrast to isolated V_S . Specifically, the onset of the VBM indicated by the STS curve obtained from the V_S (represented in Fig. 1 e) does not coincide with the onset of the VBM extracted from the EDC in Fig. 3 (b). However, the STS curve obtained from the 1D MTB (illustrated by the dark red line) exhibits the onset of the VBM that aligns with the EDC derived from the nARPES data acquired after the annealing step (depicted by the light red line).

6. The main text, and the SM, does not mention what information Fig. 4(b) provides. It looks like a TOC figure, but it was, perhaps, placed in a wrong place. The authors should either explain it or remove it from Fig. 4.

Thank you for your comment. Our intention was to create a summary figure of what we found for the correlation measurements using STM and nARPES that provide a clear picture to the reader. In addition, we are also showing the local band bending behavior in STM. We did not clearly refer to the figure in the text, which we do apologize for this error. We have now made the discussion in the figure to be more clearly integrated in the text. Fig. 4 showcases the band bending behavior as one ramps the STM tip to a

MTB on a local level. Fig. 4 (b) highlights atomic locations of the STM tip with respect to the defect and subsequently the data in Fig 4 (a). We have added the following text to better clarify this statement.

Fig. 4: **Tomonaga Luttinger Liquid Spatial Dependence.** (a) LDOS spectra collected from pristine WS₂ to an MTB ($V_{modulation} = 5$ mV, $I_{set} = 150$ pA) and then recorded in reverse. The as-measured VBM, CBM, HOS, and LUS are highlighted with relative positions to E_F . (b) Overview of the process of charge transfer from graphene into an MTB that is able to host highly correlated electron states is depicted, where the E_F of graphene is shifted and locally modified due to the presence of an MTB.

Both nARPES measurements and STS highlight the importance in choice of substrate to determine the electronic properties of 1D defects in TMDs. The process sketched in Fig. 4 (b) illustrates the reduced screening effect caused by graphene, which occurs due to the depletion of electrons in favor of MTB states that enhances electron-electron interaction within the defect. This interaction leads to a metallic behavior that ultimately influences the renormalization of the band-gap in WS₂.

REFERENCES

- [1] Zhu, T. et al. Imaging gate-tunable Tomonaga-Luttinger liquids in 1H-MoSe₂ mirror twin boundaries. *Nat. Mater.* **21**, 748 (2022).

- [2] Schuler, B. et al. Large spin-orbit splitting of deep in-gap defect states of engineered sulfur vacancies in monolayer WS₂. *Phys. Rev. Lett.* **123**, 076801 (2019).
- [3] Xia, Y. et al. Charge density modulation and the Luttinger Liquid state in MoSe₂ mirror twin boundaries. *ACS Nano* **14**, 10716 (2020).
- [4] Jolie, W. et al. Tomonaga-Luttinger liquid in a box: Electrons confined within MoS₂ mirror-twin boundaries. *Phys. Rev. X* **9**, 011055 (2019).
- [5] Meden, V. et al. Luttinger liquids with boundaries: Power-laws and energy scales. *Eur. Phys. J. B* **16**, 631 (2000).
- [6] Ryan, J. C. & Reinecke, T. L. Band-gap renormalization of optically excited semiconductor quantum wells. *Phys. Rev. B*, vol. 47, pp. 9615–9620, Apr 1993.
- [7] Wang, L. et al. Direct observation of one-dimensional Peierls-type charge density wave in twin boundaries of monolayer MoTe₂. *ACS Nano* **14**, 8299 (2020).
- [8] Peierls, R. & Peierls, R. E. *Quantum theory of solids*. (Oxford Univ. Press, 1955).
- [9] Schindler, F. et al. Higher-order topology in bismuth. *Nat. Phys.* **14**, 918 (2018).
- [10] Wu, D. et al. Uncovering edge states and electrical inhomogeneity in MoS₂ field-effect transistors. *Proc. Natl. Acad. Sci. U.S.A.* **113**, 8583 (2016).
- [11] Yang, G. et al. Possible luttinger liquid behavior of edge transport in monolayer transition metal dichalcogenide crystals. *Nat. Commun.* **11**, 659 (2020).
- [12] Kim, B. J. et al. Distinct spinon and holon dispersions in photoemission spectral functions from one-dimensional SrCuO₂. *Nat. Phys.* **2**, 397 (2006).
- [13] Mourik, V. et al. Signatures of Majorana fermions in hybrid superconductor-semiconductor nanowire devices. *Science* **336**, 1003 (2012).
- [14] Haldane, F. D. M. ‘Luttinger liquid theory’ of one-dimensional quantum fluids. I. Properties of the Luttinger model and their extension to the general 1D interacting spinless Fermi gas. *J. Phys. C: Solid State Phys.* **14**, 2585 (1981).
- [15] Fabrizio, M. & Gogolin, A. O. Interacting one-dimensional electron gas with open boundaries. *Phys. Rev. B* **51**, 17827 (1995).
- [16] Anfuso, F. & Eggert, S. Luttinger liquid in a finite one-dimensional wire with box-like boundary conditions. *Phys. Rev. B* **68**, 241301 (2003).
- [17] Kane, C., Balents, L. & Fisher, M. P. A. Coulomb interactions and mesoscopic effects in carbon nanotubes. *Phys. Rev. Lett.* **79**, 5086 (1997).
- [18] Newville, M., Stensitzki, T., Allen, D. B., & Ingargiola, A. LMFIT: Non-Linear Least-Square Minimization and Curve-Fitting for Python. <https://lmfit.github.io/lmfit-py/> (2014).
- [19] Murray, C. et al. Band bending and valence band quantization at line defects in MoS₂. *ACS Nano* **14**, 9176 (2020).
- [20] Batzill, M. Mirror twin grain boundaries in molybdenum dichalcogenides. *J. Condens. Matter Phys.* **30**, 493001 (2018).
- [21] Gibertini, M. & Marzari, N. Emergence of one-dimensional wires of free carriers in transition-metal-dichalcogenide nanostructures. *Nano Lett.* **15**, 6229 (2015).

- [22] Lehtinen, O. et al. Atomic scale microstructure and properties of Se-deficient two-dimensional MoSe₂. *ACS Nano* **9**, 3274 (2015).
- [23] Komsa, H.-P. & Krasheninnikov, A. V. Engineering the electronic properties of two-dimensional transition metal dichalcogenides by introducing mirror twin boundaries. *Adv. Electron. Mater.* **3**, 1600468 (2017).
- [24] Ma, Y. et al. Angle resolved photoemission spectroscopy reveals spin charge separation in metallic MoSe₂ grain boundary. *Nat. Commun.* **8**, 14231 (2017).

Reviewer #1:

The author addressed my concerns carefully, I think the work is interesting enough for publication.

Thank you very much for all of your help and support during the revision of this manuscript. We have very much appreciated all of the helpful feedback to help increase the quality of our work.

Reviewer #2:

The authors have revised the manuscript and provided detailed response to the Reviewers' criticism. Some of the raised issues were satisfactorily addressed, some were not.

However, I will not go through these in detail, since I realized that the manuscript contains a major flaw: Vast majority of the defects shown in the manuscript are not mirror twin boundaries (MTBs)! E.g, those in Figs. 1(b), 2(a), and 5(a) are all short segments that begin and end from pristine WS₂, which is impossible for MTBs since the lattice orientation must be mirrored in the two sides. In fact, Figs. 1(b) and 1(c) both look like the lattice orientation is not mirrored (in Fig. 1(a) I am looking at the orientation of the 3-fold symmetric vacancy features). The only case that could be MTB is in the first panel of Supplementary Fig. 9.

These line defects are instead likely vacancy lines, which would also agree with the previous work from MoS₂. Observing quantum well states and strong electronic correlations in vacancy lines could be novel. However, in order to address this issue and possibly changing the story to vacancy lines, everything in the manuscript has to be changed and therefore the present manuscript cannot be considered for publication in Nature Communications.

We greatly appreciate your thorough evaluation of our revised manuscript and your constructive feedback. The main criticism of the reviewer is that the observed line defects cannot be mirror twin boundaries because of the lack of full lattice orientation inversion. We are wondering if the issue is partly semantics, because we clearly demonstrate inversion symmetry associated with MTBs and the lack of chalcogen vacancies within the 1D defect line using ncAFM. In addition, we show clear spectroscopic signature of TLLs that have been previously shown for both types of MTB symmetries we present within WS₂. We do realize that MTBs in TMDs are typically associated with large 1D features (linear or triangular) that evoke an inversion symmetry across the entire flake, which are typically introduced during the growth process. We are demonstrating a new way to engineer these 1D defect lines post growth which exhibit the exact same inversion symmetry as MTBs, including the formation of a correlated electron system, in this case, a TLL. However, to be more clear, in terms of semantics and to avoid any confusion, we now refer to 1D inversion domains (IDs).

The main finding and critical message of this manuscript is that we show that 1D IDs are not sufficient to exhibit a TLL phase alone, and that it requires a heterostructure with graphene. We don't observe the formation of a TLL in 1D IDs when it is contacted to an underlying WS₂ layer. We realize that to convince the readers that our finding directly applies also to TLLs found in MTBs, we need to provide a clear description that shows the equivalency in terms of the structure of the 1D IDs with a traditional MTBs. We are providing a brief discussion of the main arguments to the reviewer here, which we also argue in the updated manuscript.

First and foremost, we employ, parallel to scanning tunneling spectroscopy (STS), ncAFM that provides precise insight into the local atomic structure of the feature we are observing (see updated Supplementary Fig. 4). We observe two inversion symmetries of the three that are predicted to be the most stable in the formation of 1D defects in TMDs¹⁻⁴. In our case we observe the 4I4E and 4I4P inversion symmetry (referred to AID and SID). These two types of inversion symmetries that we find experimentally are the ones found in MTBs that exhibit TLLs. We show that the electronic structure we measure via STS

(shown in Supplementary Fig. 5 below) is what has been observed and predicted for both the purely 1D and 1D 60° 4I4E structures⁵ and consistent with measurements for both the purely 1D and 1D 60° 4I4P structure⁶⁻⁸. A ncAFM methodology for atomic species identification, described by Barja et al.⁶, is used to identify both types of IDs.

We do agree with the reviewer that 1D vacancy lines would provide a fascinating playground, e.g., to study the hopping mechanism that could lead to 1D transport. However, not only have we clearly demonstrated the inversion symmetry with ncAFM and that we do not measure the presence of chalcogen vacancies, we have extended experience creating 0D and 1D defects in these systems, but we have so far not been able to align more than 4 vacancies in a true 1D line regardless of the formation mechanism (UHV annealing, laser annealing, He⁺ ion bombardment, Ar⁺ ion bombardment) and subsequent annealing protocols. Moreover, our 1D IDs are between 2-17 nm.

We believe based on the reviewers' doubts, it is important to point out that our approach to create these MTBs goes through an evolution. At first we create dense regions of both chalcogen vacancies and 1D IDs. Upon additional annealing, we find longer and longer 1D ID formations that start creating lattice-oriented zig-zag networks (as shown in Fig. 1 (b) and Supplementary Fig. 1) with a large reduction of chalcogen vacancies. Similar mechanisms were observed in TEM studies of creating 1D defects, which are introduced under ebeam exposure and immediately start to form into either MTBs or IDs^{2,9,10}. Something that is consistent across much of the literature is that elongated 1D vacancy lines form 1D IDs by means of mass transport⁹. We are glad the reviewer brought this up, as we do not want to confuse readers as many MTBs do show 60° formations (as we also show in Supplemental Fig. 9), but these formations are observed to a smaller extent in the amount of fabrication steps presented here. The authors expect 60° formations to be more prevalent with increased annealing/sputtering steps, as the formation observed in Supplementary Fig. 9 occurred after an SAA_{step} followed by an overnight anneal at 250 °C. In fact, the 1D structures have been shown to be intermediate strained phases², where the 60° formations are relaxed with little to no strain. However, again, our measurements do not indicate the formation of vacancy lines, which is expected as we are annealing during all steps of the fabrication process and, as such, providing sufficient energy for atomic migration in the system. Our imaging data would suggest some amount of strain imparted to the 1D ID defect. This, however, is difficult to quantify as our images are limited by size, number of pixels, and dwell time needed over each point for sufficient signal to noise. Additionally, any drift from slight thermal fluctuations or piezo hysteresis during data acquisition may play a role. As such, we are confident in the structure assignment (which is well-matched to the expected electronic structure), but quantifying the amount of strain in such systems would involve another complete study with extensive ncAFM combined with DFT simulations.

The exact formation of 1D IDs under the synthetic procedure we show and their subsequent evolution into 60° MTBs is very interesting and we are currently preparing a follow-up manuscript on this topic, since it goes beyond the scope of this paper. The key message of this manuscript is that a 1D inversion symmetry found also found in MTBs is not sufficient to create a TLL. It is the heterostructure and direct contact with graphene that is necessary to create the local quantum phase transition into a correlated electron system (i.e., a TLL) that does not require ultra low temperatures. In following we highlight the changes in the manuscript that make the discussion on what exact 1D system we are studying and effects much clearer to the reader.

Supplementary Fig. 4: **Atomic Force Imaging.** (a) Constant current image over an isolated AID defect ($I_{tunnel} = 30$ pA, $V_{sample} = 1.2$ V). Scale bar, 1.5 nm. (b) ncAFM with a CO functionalized tip ($V_{sample} = 0.0$ V) collected over the same AID with a ball model showing atomic locations. Scale bar, 0.4 nm. (c) A linescan over the red region shown in (b) that depicts two types of sulfur measured, where one is assigned to the top position (yellow) and the other to the bottom position (cyan) where metal sites appear as dips in frequency shift images. Nearby depressions near the AID reflect oxygen atom (circled in orange) chalcogen substitutions within an otherwise unmodified WS_2 lattice. The ID symmetry is shown schematically in (d) for clarity, where both types of sulfur, metal, and hollow sites are distinguishable in ncAFM measurements. A second isolated ID is shown in (e) that is identified to be an SID ($I_{tunnel} = 30$ pA, $V_{sample} = 1.2$ V). Scale bar, 1.5 nm. (f) ncAFM confirms the location of top sulfur, tungsten, and hollow sites ($V_{sample} = 0.0$ V), where a linescan of across the SID in (g) shows the location of dual tungsten surrounded by sulfur sites which is also shown schematically in (h). Inversion unit cells are shown in bold red in both (b) and (f).

Supplementary Fig. 5: **LDOS Mapping**. Conductance maps ($V_{modulation} = 5$ mV) performed across an SID show dual-line orbital behavior at (a) 0.132 eV (LUS) and (b) -0.025 eV (HOS) that is spatially out of phase. (c) Accumulated conductance maps across a single-line of the SID ($V_{modulation} = 5$ mV) are further shown as a function of bias, where the number of nodes increase as bias voltage is increased. Scale bars, 1 nm. Peak assignments can be made by solving for local maxima along a line profile, which is compiled in (d). Conductance maps (dI/dV) of the as-measured (e) 0.047 eV (LUS) and (f) -0.047 eV (HOS) that are spatially in phase within an AID. (g) Compiled conductance maps ($V_{modulation} = 5$ mV) across the single-line AID are further shown, where the number of nodes decreases as the voltage is increased. (h) Line profiles extracted from (g) detailing local maxima.

Page 2:

Recently, this phenomenon has also been shown to exist within 1D defects in two-dimensional (2D) TMDs at a plane of lattice points where the crystal structures on either side of the interface are mirrored, which defines a mirror twin boundary (MTB) within monolayer (ML) TMDs^{5,7,8}, where similar symmetries along a single line have also been referred to as inversion domains (IDs)^{2,9,11}. IDs (and MTB formations) have been predicted to exhibit metallic properties and to form out of sub-stoichiometric metal ($M = \text{Mo}, \text{W}$) or from depleted chalcogen ($X = \text{S}, \text{Se}, \text{Te}$) in MX_2 materials^{1,3,4,6}. Additionally, 1D chalcogen vacancy lines have been shown to form into IDs during mass transport under thermal annealing^{9,10}.

Page 3:

Chalcogen vacancies and vacancy lines, which have been measured in TMDs¹², have been shown to reorganize into 1D defect formations and subsequent IDs at sufficient annealing temperatures or by ion beam activation during imaging^{2,9,10}.

Reviewer #3:

I've carefully read the authors' reply and the revised manuscript. I appreciate with the authors' efforts for improving it. The authors managed to clarify most of my concerns and to strengthen the importance and significance of the manuscript. I thus recommend it for publication in Nature Communications.

We are very grateful for your recommendation and appreciate that you have read and understood everything presented in our work. Thank you so much for your insight and assistance in bringing this manuscript to the next level.

REFERENCES

- [1] Lehtinen, O. et al. Atomic scale microstructure and properties of Se-deficient two-dimensional MoSe_2 . *ACS Nano* **9**, 3274 (2015).
- [2] Yan, A. et al. Dynamics of symmetry-breaking stacking boundaries in bilayer MoS_2 . *J. Phys. Chem. C*, **121**, 22559 (2017).
- [3] Gibertini, M. & Marzari, N. Emergence of one-dimensional wires of free carriers in transition-metal-dichalcogenide nanostructures. *Nano Lett.* **15**, 6229 (2015).
- [4] Batzill, M. Mirror twin grain boundaries in molybdenum dichalcogenides. *J. Condens. Matter Phys.* **30**, 493001 (2018).
- [5] Jolie, W. et al. Tomonaga-Luttinger liquid in a box: Electrons confined within MoS_2 mirror-twin boundaries. *Phys. Rev. X* **9**, 011055 (2019).
- [6] Barja, S. et al. Charge density wave order in 1D mirror twin boundaries of single-layer MoSe_2 . *Nat. Phys.* **12**, 751 (2016).
- [7] Xia, Y. et al. Charge density modulation and the Luttinger Liquid state in MoSe_2 mirror twin boundaries. *ACS Nano* **14**, 10716 (2020).
- [8] Zhu, T. et al. Imaging gate-tunable Tomonaga-Luttinger liquids in 1H- MoSe_2 mirror twin boundaries. *Nat. Mater.* **21**, 748 (2022).
- [9] Wang, X.-W. et al. Mass transport induced structural evolution and healing of sulfur vacancy lines and Mo chain in monolayer MoS_2 . *Rare Met.* **41**, 333 (2022).

- [10] Lin, J., Pantelides, S. T., & Zhou, W. Vacancy-induced formation and growth of inversion domains in transition-metal dichalcogenide monolayer. *ACS Nano* **9**, 5189 (2015).
- [11] Nalin Mehta, A. et al. Grain-boundary-induced strain and distortion in epitaxial bilayer MoS₂ lattice. *J. Phys. Chem. C* **124**, 6472 (2020).
- [12] Fang, S. et al. Atomic electrostatic maps of 1d channels in 2d semiconductors using 4d scanning transmission electron microscopy. *Nat. Commun.* **10**, 1127 (2019).

Reviewer #2:

Below are the comments from the reviewer's earlier report (paragraphs starting with »), the authors' response (starting with >), and the reviewer's response (starting with .). Our comments are then addressed after this below.

» The authors have revised the manuscript and provided detailed response to the Reviewers' criticism.

» Some of the raised issues were satisfactorily addressed, some were not. However, I will not go through these in detail, since I realized that the manuscript contains a major flaw: Vast majority of the defects shown in the manuscript are not mirror twin boundaries (MTBs)! E.g, those in Figs. 1(b), 2(a), and 5(a) are all short segments that begin and end from pristine WS₂, which is impossible for MTBs since the lattice orientation must be mirrored in the two sides. In fact, Figs. 1(b) and 1(c) both look like the lattice orientation is not mirrored (in Fig. 1(a) I am looking at the orientation of the 3-fold symmetric vacancy features). The only case that could be MTB is in the first panel of Supplementary Fig. 9.

» These line defects are instead likely vacancy lines, which would also agree with the previous work from MoS₂. Observing quantum well states and strong electronic correlations in vacancy lines could be novel. However, in order to address this issue and possibly changing the story to vacancy lines, everything in the manuscript has to be changed and therefore the present manuscript cannot be considered for publication in Nature Communications.

> We greatly appreciate your thorough evaluation of our revised manuscript and your constructive feedback. The main criticism of the reviewer is that the observed line defects cannot be mirror twin boundaries because of the lack of full lattice orientation inversion. We are wondering if the issue is partly semantics, because we clearly demonstrate inversion symmetry associated with MTBs and the lack of chalcogen vacancies within the 1D defect line using ncAFM. In addition, we show clear spectroscopic signature of TLLs that have been previously shown for both types of MTB symmetries we present within WS₂. We do realize that MTBs in TMDs are typically associated with large 1D features (linear or triangular) that evoke an inversion symmetry across the entire flake, which are typically introduced during the growth process. We are demonstrating a new way to engineer these 1D defect lines post growth which exhibit the exact same inversion symmetry as MTBs, including the formation of a correlated electron system, in this case, a TLL. However, to be more clear, in terms of semantics and to avoid any confusion, we now refer to 1D inversion domains (IDs).

. No, there is no semantics. It is simply impossible to have finite-length 1D defect with inversion symmetry on the two sides. Where is the inversion symmetry disappearing at the end of the 1D defect? The lattice cannot transform from "normal" to "inverted" orientation without defects. Simply try to create an atomic model for a finite length MTB segment (based on the infinite MTB structure shown in Fig. 4) and you will quickly realize this.

Thank you for your comment and for clarifying your concern. The reviewer is correct, and we certainly agree, that the lattice cannot simply transform from 'normal' to 'inverted' orientation without defects or increased strain at the end of the 1D defect line. This has already been highlighted by Lin et al., where the formation of inversion-like domain (ID) defects occurs at sufficient vacancy concentrations and is favorable over a vacancy line^{1,2}. The reconstruction happens for defects at least 8 unit cells in length (~ 2.5 nm) under electron irradiation during STEM, where all 1D metallic defects shown in our work (and that exhibit TLL behavior) are on the order of 10 nm. These 1D linear defects were shown by STEM to contract the surrounding Mo lattice and lead to defects that are highly strained, which the authors believe a number of defects could form at the end of 1D inversion-like grains via mass transport to enable strain relaxation^{1,3}. To that end, we precisely show defects and strain at the end of an isolated 1D ID in our most recent round of data (Supplementary Figure 4). Under consistent electron irradiation, the linear 1D grain boundaries begin to transform into zig-zag and triangular structures that invoke a symmetry across the

entire flake^{1,2}. This is consistent with our work, where after multiple cycles of annealing and chalcogen depletion steps (Ar^+ sputtering), we begin to see the zig-zag structures and triangular structures as shown in Supplementary Figures 7 and 9. We show that the electronic structure and LDOS is equivalent across the isolated 1D IDs, zig-zag, and triangular structures. Imaging the zoo of possible formations at defect endpoints during the growth process is indeed an interesting topic and is very much worthy of a separate investigation, which we plan to address in future studies. We clearly see and show that isolated 1D IDs, zig-zag, and triangular structure show the exact same electronic structure and form a TLL. Our focus and key finding for this manuscript remains the formation of a TLL and the critical heterostructuring with an underlying graphene substrate, which we find quite compelling. We were also able to collect additional data after the first revision showing the lack of TLL formation in 1D defects if there is contact to underlying layer of WS_2 . To best address the reviewer's concerns and to avoid detracting from the key findings, we instead focus on the electronic component and report the 1D defects as inversion-like domains (IDs), where the results do indicate formation of true MTBs after additional cycles of sputtering and annealing (Supplementary Figures 7 and 9 shown again below). Instead, the purely 1D defects that end without a triangular domain or do not go across the entire flake are strained formations that end in defects.

Supplementary Fig. 4 new: **1D ID Defect Line Mapping.** (a) Constant current image over an isolated 1D ID defect ($I_{\text{tunnel}} = 30 \text{ pA}$, $V_{\text{sample}} = 1.2 \text{ V}$). Scale bar, 1 nm. (b) Zoomed in image that is highlighted in (a) with a red box, where sequential (c) ncAFM with a CO functionalized tip ($V_{\text{sample}} = 0.0 \text{ V}$) collected over the same 1D ID showcase local structure. Scale bars, 1 nm. A chalcogen depletion defect (blue) is located at the edge of the strained 1D ID, which then forms into an SID (green).

Supplementary Fig. 7: **Differential Conductance Mapping.** dI/dV mapping ($V_{modulation} = 5$ mV) over the spectra region shown for an AID on a jet color scale, where the energy is ramped from near the VBM of WS_2 by 0.05 V to the HOS, and then from the LUS to the CBM of WS_2 . The defect imaged is shown to the upper left, where the partial portion of the mapped defect is highlighted in dashed red ($I_{tunnel} = 30$ pA, $V_{sample} = 1.2$ V). Scale bar, 4 nm. Orbitals of the as-formed TLL can be visualized on as-acquired data as a function of bias voltage, where arrows indicate decreasing bias.

Supplementary Fig. 9: **Constant-Current Density of States Decay Across an ID.** A power-law dependence is measured across a multiple ID defects on both monolayer WS₂ and multilayer (3ML) WS₂ on MLG/SiC(0001). In each image depicted, multiple fits were taken across defects shown, where representative linescans beginning at a starred point and along a given line (red) are shown with corresponding constant-current profiles with subsequent exponential fittings. Across 14 lineprofiles and 10 defects, multilayer WS₂ defects show an absolute power-law exponential parameter of 0.99 ± 0.15 , which indicates Fermionic behavior, and monolayer WS₂ defects yield a parameter of 0.49 ± 0.16 that matches expected behavior of a Luttinger liquid. Scale bars, 2nm. Fittings were performed using the lmfit package in Python⁴.

. Because of this it is also obvious why all previous research report MTBs “that evoke an inversion symmetry across the entire flake” (or large triangular domains). It’s not because the growth process happens to lead to long MTBs between inverted domains, but because that’s the only way to connect inverted domains; by going all the way from edge to edge.

The reviewer makes a claim that all previous reports of MTBs evoke an inversion symmetry across the entire flake, however inversion-like 1D grains have been observed by STEM that form after sufficient chalcogen depletion is present, which is discussed in further detail in our first comment. The reviewer is correct that a domain across the entire flake (or triangular domain) is more stable (and intuitive) due to structural arguments. However, as mentioned in our earlier comments, highly strained isolated inversion-like domains have been observed in the literature, that can end in defects.

> The main finding and critical message of this manuscript is that we show that 1D IDs are not sufficient to exhibit a TLL phase alone, and that it requires a heterostructure with graphene. We don't observe the formation of a TLL in 1D IDs when it is contacted to an underlying WS₂ layer. We realize that to convince the readers that our finding directly applies also to TLLs found in MTBs, we need to provide a clear description that shows the equivalency in terms of the structure of the 1D IDs with a traditional MTBs. We are providing a brief discussion of the main arguments to the reviewer here, which we also argue in the updated manuscript.

. That may be true for whatever this defect is, but it is not MTB (nor 1D ID).

Although we have collected additional data, which satisfies the reviewer's comment that a defect is required to host an isolated ID, we wish to underline that the primary objective of our manuscript was to investigate the electronic properties of what we have now termed inversion-like domains. Our research notably highlights the formation of a TLL phase in these structures, with a specific focus on the influence of the substrate in this context. Our findings indicate a distinct absence of TLL formation in IDs when interfaced with a WS₂ layer, contrasting with their combination with graphene. This difference underscores the critical role of the heterostructure environment in such systems. While we concur that our results showcase the strained 1D inversion-like defect, we believe our study provides valuable insights into the behavior of 1D electronic systems, as they show the same electronic behavior as the zig-zag or triangular grain boundaries, and highlights their interactions with underlying substrates. Our research contributes to a better understanding of these phenomena and lays a foundation for further investigation.

Many of the defect structures formed after SA_{step} and SAA_{step} are finite 1D IDs. The defects are strained and can form, e.g., vacancy defects at either end of the defect to relax the system (see Supplementary Fig. 4), in agreement with what has been reported in great detail for similar systems¹.

> First and foremost, we employ, parallel to scanning tunneling spectroscopy (STS), ncAFM that provides precise insight into the local atomic structure of the feature we are observing (see updated Supplementary Fig. 4). We observe two inversion symmetries of the three that are predicted to be the most stable in the formation of 1D defects in TMDs^{1–4}. In our case we observe the 4|4E and 4|4P inversion symmetry (referred to AID and SID). These two types of inversion symmetries that we find experimentally are the ones found in MTBs that exhibit TLLs. We show that the electronic structure we measure via STS (shown in Supplementary Fig. 5 below) is what has been observed and predicted for both the purely 1D and 1D 60° 4|4E structures⁵ and consistent with measurements for both the purely 1D and 1D 60° 4|4P structure^{6–8}. A ncAFM methodology for atomic species identification, described by Barja et al.⁶, is used identify both types of IDs.

. First, in many of the STM/ncAFM images shown in the manuscript, the image has been cropped in such a way that it is impossible for me to judge whether it is a true MTB or a finite-length 1D defect. Some of these defects might be one and some might be the other.

The STM and ncAFM data presented are over regions where we were able to obtain high resolution atomic mapping over the defect structure, and thus may appear 'cropped', but in fact it is the region of acquired measurement. As the electronic component of defects within WS₂ can be larger than their structural component⁵, ncAFM data were collected to highlight the atomic arrangement of the defect. The recent round of data highlights defects that occur at the end of isolated 1D IDs.

. Second, while the structure in Supplementary Fig. 4 indeed looks similar to MTBs reported in the literature, it does not show inversion symmetry. Looking at the yellow and green dots, the lattice orientation is the same, and the red triangle is drawn wrong (the right side is missing yellow dot that would fall inside the triangle).

The atomic assignments have been rechecked by the authors, however as the system is in fact strained to enable inversion-like formation, the authors have removed the red triangles in the newly updated Supplementary Fig. 3.

Supplementary Fig. 3 new: **Atomic Force Imaging.** (a) Constant current image over an isolated AID defect ($I_{tunnel} = 30 \text{ pA}$, $V_{sample} = 1.2 \text{ V}$). Scale bar, 1.5 nm. (b) ncAFM with a CO functionalized tip ($V_{sample} = 0.0 \text{ V}$) collected over the same AID with a ball model showing atomic locations. Scale bar, 0.4 nm. (c) A linescan over the red region shown in (b) that depicts two types of sulfur measured, where one is assigned to the top position (yellow) and the other to the bottom position (cyan) where metal sites appear as dips in frequency shift images. Nearby depressions near the AID reflect oxygen atom (circled in orange) chalcogen substitutions within an otherwise unmodified WS_2 lattice. The ID symmetry is shown schematically in (d) for clarity, where both types of sulfur, metal, and hollow sites are distinguishable in ncAFM measurements. A second isolated ID is shown in (e) that is identified to be an SID ($I_{tunnel} = 30 \text{ pA}$, $V_{sample} = 1.2 \text{ V}$). Scale bar, 1.5 nm. (f) ncAFM confirms the location of top sulfur, tungsten, and hollow sites ($V_{sample} = 0.0 \text{ V}$), where a linescan of across the SID in (g) shows the location of dual tungsten surrounded by sulfur sites which is also shown schematically in (h).

> We do agree with the reviewer that 1D vacancy lines would provide a fascinating playground, e.g., to study the hopping mechanism that could lead to 1D transport. However, not only have we clearly demonstrated the inversion symmetry with ncAFM and that we do not measure the presence of chalcogen vacancies, we have extended experience creating 0D and 1D defects in these systems, but we have so far not been able to align more than 4 vacancies in a true 1D line regardless of the formation mechanism (UHV annealing, laser annealing, He⁺ ion bombardment, Ar⁺ ion bombardment) and subsequent annealing protocols. Moreover, our 1D IDs are between 2-17 nm.

. To be clear, I am not claiming that it is 1D vacancy line. It was just the first candidate that came to mind based on previous literature.

We appreciate the reviewer's clarification and hope we have made the manuscript and our findings overall more clear to future readers.

> We believe based on the reviewers doubts, it is important to point out that our approach to create these MTBs goes through an evolution. At first we create dense regions of both chalcogen vacancies and 1D IDs. Upon additional annealing, we find longer and longer 1D ID formations that start creating lattice-oriented zig-zag networks (as shown in Fig. 1 (b) and Supplementary Fig. 1) with a large reduction of chalcogen vacancies.

. And these I would agree look like MTBs since each domain appear to be bounded by MTBs from all sides, and MTBs are connected by local defective regions.

Thank you for your comments, where local defective regions at the edge were also found in recent data highlighting an isolated linear inversion-like domain.

> Similar mechanisms were observed in TEM studies of creating 1D defects, which are introduced under ebeam exposure and immediately start to form into either MTBs or IDs^{2,9,10}. Something that is consistent across much of the literature is that elongated 1D vacancy lines form 1D IDs by means of mass transport⁹. We are glad the reviewer brought this up, as we do not want to confuse readers as many MTBs do show 60° formations (as we also show in Supplemental Fig. 9), but these formations are observed to smaller extent in the amount of fabrication steps presented here. The authors expect 60° formations to be more prevalent with increased annealing/sputtering steps, as the formation observed in Supplementary Fig. 9 occurred after an SAAstep followed by an overnight anneal at 250 °C. In fact, the 1D structures have been shown to be intermediate strained phases², where the 60° formations are relaxed with little to no strain. However, again, our measurements do not indicate the formation of vacancy lines, which is expected as we are annealing during all steps of the fabrication process and, as such, providing sufficient energy for atomic migration in the system. Our imaging data would suggest some amount of strain imparted to the 1D ID defect. This, however, is difficult to quantify as our images are limited by size, number of pixels, and dwell time needed over each point for sufficient signal to noise. Additionally, any drift from slight thermal fluctuations or piezo hysteresis during data acquisition may play a role. As such, we are confident in the structure assignment (which is well-matched to the expected electronic structure), but quantifying the amount of strain in such systems would involve another complete study with extensive ncAFM combined with DFT simulations.

> The exact formation of 1D IDs under the synthetic procedure we show and their subsequent evolution into 60° MTBs is very interesting and we are currently preparing a follow-up manuscript on this topic, since it goes beyond the scope of this paper. The key message of this manuscript is that a 1D inversion symmetry found also found in MTBs is not sufficient to create a TLL. It is the heterostructure and direct contact with graphene that is necessary to create the local quantum phase transition into a correlated electron system (i.e., a TLL) that does not require ultra low temperatures. In following we highlight the changes in the manuscript that make the discussion on what exact 1D system we are studying and effects much clearer to the reader.

. In conclusion, some of the presented results might correspond to MTB, whereas the images showing clear finite length certainly do not. This is what I stated in my previous report and since the authors have not taken this into account in the revised manuscript, it still cannot be considered for publication in Nature Communications.

The reviewer again claims that finite length mirror twin boundaries are not possible. However, both Lehtinen et al. and Lin et al. have shown that 1D vacancy lines form into inversion-like domains which have been shown to be highly strained in the isolated 1D form^{1,2}. Our data would agree with these works and the multitude of others, where 1D chalcogen depletion eventually forms into mirror twin boundaries either across the entire flake or in triangular formations. This is in fact what we visualize in our experiments, where increased cycles of Ar⁺ sputtering and annealing lead to the formation of elongated zig-zag structures or the triangular formations. Additionally, the most recent data show that not only inversion-like domains form with finite length as shown by Lin et al., but also that these defects end in defective regions, as the reviewer mentioned in an earlier comment, "And these I would agree look like MTBs since each domain appear to be bounded by MTBs from all sides, and MTBs are connected by local defective regions." Again, the key message we wish to convey is that regardless of the exact structure at the end of 1D IDs or within fully-formed IDs that relax strain across the entire flake, they show the same electronic structure and subsequent metallic behavior. This 1D behavior, which shows the signature of a TLL, is enabled by the underlying of the substrate, where screening and stabilization by graphene determines the stability of the quasi-particle excitation. We hope that the latest round of data and revisions help alleviate any concerns for publication in Nature Communications.

Additionally, where isolated 1D IDs are strained, they have been shown to form out of chalcogen vacancies, and eventually form into zig-zag (see Supplementary Fig. 7) and 60° (see Supplementary Fig. 9) IDs that reduce strain across the WS₂ flake^{1,2}. This is also observed in the presented work.

REFERENCES

- [1] Lin, J., Pantelides, S. T., & Zhou, W. Vacancy-induced formation and growth of inversion domains in transition-metal dichalcogenide monolayer. *ACS Nano* **9**, 5189 (2015).
- [2] Lehtinen, O. et al. Atomic scale microstructure and properties of Se-deficient two-dimensional MoSe₂. *ACS Nano* **9**, 3274 (2015).
- [3] Wang, X.-W. et al. Mass transport induced structural evolution and healing of sulfur vacancy lines and Mo chain in monolayer MoS₂. *Rare Met.* **41**, 333 (2022).
- [4] Newville, M., Stensitzki, T., Allen, D. B., & Ingargiola, A. LMFIT: Non-Linear Least-Square Minimization and Curve-Fitting for Python. <https://lmlfit.github.io/lmlfit-py/> (2014).
- [5] Schuler, B. et al. Large spin-orbit splitting of deep in-gap defect states of engineered sulfur vacancies in monolayer WS₂. *Phys. Rev. Lett.* **123**, 076801 (2019).

Reviewer 2 (Remarks to the Author):

The revised version of the manuscript contains no major changes.

In this latest version, we have performed significant revisions, including multiple follow-up experiments that have led to the identification of 1D chalcogen vacancy lines, as you originally suspected. Using a combination of ncAFM and STM/STS, we have expanded our analysis to include the structure and electronic properties of MTB intermediates, fully formed MTBs, and vacancy lines.

While we value the importance of characterizing these defects, we would like to emphasize that the core focus of this manuscript lies in demonstrating the pivotal role of graphene as a substrate in enabling correlated electronic states across various 1D defect types in WS₂. The additional experiments and revisions have strengthened our ability to present this broader and more impactful message.

We thank you again for your detailed observations, which have contributed to refining the clarity and scope of our work.

The authors claim that the isolated 1D defects are mirror twin boundaries (MTB) or something the authors called "inversion-like domain" (ID). As I wrote in my previous report, this is obviously not possible. I asked the authors to provide a model for their 1D ID/MTB, but they did not. I hoped that had they attempted they would have realized it is not possible.

After extensive additional experiments since our last revision we have identified chalcogen vacancy lines (in updated Supplementary Fig. 4) and have measured additional intermediate 4|4P-like and 4|4E-like formations (Supplementary Fig. 5). Lin et al. and Wang et al. were able to observe metal chains form before the observation of a 4|4P-like structure (shown below)^{1,2}. Lin et al. was able to observe 4|4E-like structures, at a single defect end point, that evolved from an isolated sulfur vacancy line. This defect was referred to as a 4|4E GB-like structure. Under constant electron irradiation, these defects then turn into the more stable, across-the-flake or 60°, MTB formations. This data shows that complex intermediate MTB formations are possible and are further supported by the evidence gathered in our measurements. The authors agree that the formation of isolated fully formed MTBs are likely not feasible (and would be energetically unfavorable), but intermediate formations that may be relaxed by additional defects (such as chalcogen vacancies or metal chains) are possible.

The reviewer originally stated that "These line defects are instead likely vacancy lines, which would also agree with the previous work from MoS₂. Observing quantum well states and strong electronic correlations in vacancy lines could be novel. However, in order to address this issue and possible changing the story to vacancy lines, everything in the manuscript has to be changed and therefore the present manuscript cannot be considered for publication in Nature Communications." At the time of this earlier revision, the authors did not have supporting data for this well appreciated comment. This is driven by the fact that our ncAFM data are low throughput, due to the reactive nature of the defective substrate with CO functionalized tips, and that our measurements are inherently local. The authors now have clear data supporting this suggestion and the authors have revised the manuscript accordingly.

Supplementary Fig. 4: **Atomic Force Imaging.** (a) Constant current image over an isolated 4|4E-like 1DM defect ($I_{tunnel} = 30$ pA, $V_{sample} = 1.2$ V). Scale bar, 1.5 nm. (b) nAFM ($V_{sample} = 0.0$ V) collected over the same 4|4E-like 1DM with a ball model showing atomic locations. Scale bar, 0.4 nm. (c) A linescan over the red region shown in (b) that depicts two types of sulfur measured, where one is assigned to the top position (yellow) and the other to the bottom position (cyan) where metal sites appear as dips in frequency shift images. Depressions near the 4|4E-like 1DM reflect oxygen atom (circled in orange) chalcogen substitutions within an otherwise unmodified WS_2 lattice. A structure model is shown schematically in (d). A second isolated 1DM is shown in (e) that is identified to be an 4|4P-like 1DM ($I_{tunnel} = 30$ pA, $V_{sample} = 1.2$ V). Scale bar, 1.5 nm. (f) nAFM confirms the location of top sulfur, tungsten, and hollow sites ($V_{sample} = 0.0$ V), where a linescan across the 4|4P-like 1DM in (g) shows the location of dual tungsten surrounded by sulfur sites that is also shown schematically in (h). Scale bar, 0.4 nm. A 1DM chalcogen vacancy line is shown in (i) ($I_{tunnel} = 30$ pA, $V_{sample} = 1.2$ V). Scale bar, 1.5 nm. (j) Sulfur vacancies are measured in nAFM ($V_{sample} = 0.0$ V). A linescan across the 1DM in (k) shows the tungsten and lower sulfur sites, schematically depicted in (l). Scale bar, 0.4 nm.

Supplementary Fig. 5: 1DM Defect End Point Mapping. (a) Constant current image over an isolated 1DM defect ($I_{tunnel} = 30$ pA, $V_{sample} = 1.2$ V) terminating in an end point defect. Scale bar, 1 nm. (b) Zoomed in image that is highlighted in (a) with a red box, where sequential (c) ncAFM with a CO functionalized tip ($V_{sample} = 0.0$ V) collected over the same 1DM showcase local structure. Scale bars, 1 nm. A chalcogen depletion region (blue) is located at the edge of the strained 1DM, which then forms into an isolated strained 4|4P-like structure (green). Chalcogen (sulfur) sites are highlighted with yellow spheres and metal (tungsten) sites are highlighted with green spheres. (d) Another defect is measured that appears as an intermediate formation before a fully relaxed MTB ($I_{tunnel} = 30$ pA, $V_{sample} = 1.2$ V). Scale bar, 1 nm. ncAFM over an endpoint region (highlighted with a red box) shows strained metallic-rich regions that form into a 4|4P-like structure.

We make use of ncAFM with a functionalized CO tip to identify chalcogen vacancy lines, MTB-like defects, and chalcogen vacancy structures combined with MTB-like defects (see Supplementary Figs. 4 and 5). These defects are characterized by their distinct electronic behaviors—most are tungsten rich and exhibit electron-like characteristics, while sulfur rich 1DMs demonstrate hole-like behavior (see Supplementary Fig. 6). This classification is based on the modulation of electron density with bias^{3,4}. Notably, the band dispersion observed in these defects mirrors that of known MTB structures^{5–7}. For the hole-like behavior, our findings align with the 4|4E MTB similar to what was reported by Jolie et al., and for the electron-like modulation, we observe this behavior in all tungsten rich defects that include chalcogen vacancy lines and symmetries resembling the 4|4P MTB found by Zhu et al. All structures display features of a TLL^{3,4,8}. In addition to defect structures measured with ncAFM, we also show a power-law dependence in as-acquired STM images in single-line defects and fully formed 60° MTB formations, where this behavior (characteristic of a TLL) is not present in structures that instead contact an underlying layer of WS₂ (see Supplementary Fig. 7). An isolated 1D MTB in an otherwise unaltered MX₂ lattice would be highly strained. If isolated MTB-like structures are formed, additional defects form (or combine) to help relax the lattice, such as Mo chains or chalcogen vacancies, at either the endpoints or within 1DMs, as demonstrated in Supplementary Fig. 5. This observation highlights a defect at the 1DM endpoint, where we measured a strained 4|4P-like symmetry thereafter. Our analysis reveals that among the types of 1DM defects measured, the isolated MTB-like structures resemble intermediate defects in the process of forming fully relaxed MTB structures from V_S defects^{1,2,9}. Chalcogen vacancy lines,

however, create less lattice strain of the isolated 1D defects measured with a single SAA_{step} cycle, and likely represent the majority of defects created.

Wang et al. Fig. S10: **The atomic structures of underlying S vacancy lines.** a, the atomic structure of three rows of S vacancy lines before the formation of Mo chains in Figure S9b. b, the atomic structure of inversion domain boundary before the formation of Mo chains in Figure S9c. c, the atomic structure of Mo₂S₃ before the formation of Mo chains in Figure S9e, f, i.

Lin et al. Fig. 2: **Agglomeration of selenium vacancies into line defects in monolayer MoSe₂.** (a) STEM Z-contrast image of the SL line defect with the DFT-optimized structure overlaid. Inset: Side view of the structure model. The nearby Se₂ columns are slightly misaligned. (b) STEM Z-contrast image of the 4/4E GB-like structure (highly strained 4/4E 60° GB) evolved from the SL line defect, which contains deformed strings of four-fold rings. The white dashed lines indicate the centers of these defects, and the yellow dashed arrows highlight the bond length of the Mo sublattice in the defect regions. Scale bars: 0.5 nm.

The authors try to support their claims by citing literature in rebuttal letter (and in manuscript): "The reviewer again claims that finite length mirror twin boundaries are not possible. However, both Lehtinen et al. and Lin et al. have shown that 1D vacancy lines form into inversion-like domains which have been shown to be highly strained in the isolated 1D form^{1,2}. "

There is no "isolated 1D form" in either Lehtinen et al or Lin et al. In both works, the finite length defect is always a vacancy line. The MTBs generated from the vacancy lines are always triangular loops. If not generated from vacancies (chalcogen deficiency), the MTBs cross the whole sample as grain boundaries.

We apologize for not being explicit in our previous reply. The comment from Lin et. is explained above. Our comment from Lehtinen et al. was from the fact that sulfur vacancy lines showed higher formation energy compared to MTB structures (Lehtinen et al. Figure 2). Please refer to the first comment for more detail. Lin et al. does show an isolated 4/4E GB-like intermediate as shown above in Lin et al. Figure 2. Additionally, Wang et al. shows metal chain formations (that are also metallic in nature) forming at

the end point of a 4|4P structure. The authors anticipate that defects at end points enable some level of stabilization of the as-measured, and strained, 4|4E-like and 4|4P-like structures.

Normal or shear strain does not help in achieving lattice inversion. Disclination-induced 60 degree bend of the lattice should be easy to confirm in the STM/ncAFM images, but there is no indications of this (and has never been reported previously).

After successive process treatments, 60-degree broken (and merged) bend formations were measured (shown in newly updated Fig. 2, Supplementary Fig. 8, and Supplementary Fig. 12). Additionally, only chalcogen vacancy line formations showed 120-degree bend formations within our measurements. Importantly, all of these defects showed the capacity to form a TLL with matching electronic structure (Supplementary Fig. 12).

Fig. 2: Dense LDOS Mapping. (a) An electron-like 1DM is measured topographically, where the red line shown depicts a length of 18.96 nm ($I_{tunnel} = 20$ pA, $V_{sample} = 1.4$ V). Dense local density of states spectra is collected along (a) an electron-like 1DM (1x128x400 pixels) ($V_{modulation} = 5$ mV, $I_{set} = 150$ pA) beginning at the starred point along the red line. This is shown as a (b) function of bias and distance. The HOS and LUS are identified (c) at -0.032 V (orange) and 0.025 V (green). A 1D particle-in-a-box behavior is present above and below the HOS and LUS (maxima labeled as starred points). The number of nodes (N) increases linearly (N+1) from the LUS to LUS+1 (8 to 9), and also decreases from the HOS to HOS-1 (8 to 7). An FFT of (b) is shown in (d), where a spin and charge separation onset is seen above and below the E_F , and both the spin (blue) and charge (red) branches can be monitored from -0.15 V to 0.15 V. A K_c value of 0.47 is extracted from (d).

Supplementary Fig. 8: **Electron-like 1DM Dispersion Length Dependence.** Electron-like dispersions are collected on 1DMs with different lengths. Constant current images over 1DMs that are (a) 11.19 nm ($I_{tunnel} = 30$ pA, $V_{sample} = 1.2$ V), (b) 14.86 nm ($I_{tunnel} = 30$ pA, $V_{sample} = 1.4$ V), and (c) 20.44 nm ($I_{tunnel} = 80$ pA, $V_{sample} = 1.2$ V) are collected. Scale bars, 2 nm. Dense scanning tunneling spectra ($V_{modulation} = 5$ mV, $I_{set} = 150$ pA) are collected along the red line, beginning at the starred point. Results are shown for the (d) 11.19 nm defect (1x128x500 pixels), (e) 14.86 nm defect (1x128x400 pixels), and the (f) 20.44 nm defect (1x128x400 pixels). Using the HOS as a reference, a node spacing of is measured.

Supplementary Fig. 12: **Electronic Structure Comparison of Different 1DMs.** Constant current image over (a) a fully relaxed MTB across WS_2 edge regions ($I_{\text{tunnel}} = 30 \text{ pA}$, $V_{\text{sample}} = 1.2 \text{ V}$), (b) a vacancy line structure showing 120 degree rotation within WS_2 ($I_{\text{tunnel}} = 20 \text{ pA}$, $V_{\text{sample}} = 1.4 \text{ V}$), (c) a 4|4E-like structure ($I_{\text{tunnel}} = 30 \text{ pA}$, $V_{\text{sample}} = 1.2 \text{ V}$), and (d) a 4|4P-like 1DM ($I_{\text{tunnel}} = 30 \text{ pA}$, $V_{\text{sample}} = 1.2 \text{ V}$). Scale bars, 2 nm. (e) dI/dV spectra recorded over each 1DM defect case, where all show a measurable E_{gap} ($V_{\text{modulation}} = 5 \text{ mV}$) near the E_F . In addition, we highlight dI/dV linescans extracted from the HOS as a function of defect distance (f-i), which all exhibit similar oscillatory behavior.

"Our data would agree with these works and the multitude of others, where 1D chalcogen depletion eventually forms into mirror twin boundaries either across the entire flake or in triangular formations."

And all those works show that the intermediate phase is vacancy line. If the results are consistent with previous works, then the defects should be vacancy lines.

The authors apologize that our previous measurements missed this important defect formation and hope that this is clearly reflected in the updated main text and supplementary information. Additionally, we hope that we have expressed the complex nature of intermediate defect formation and plan to address this more directly in separate experiments on similar systems.

In the end, I want to note that vacancy lines and mirror twin boundaries have very different structure and properties and they cannot be mixed or grouped into a general "1D defect".

Thank you again for your helpful insight. We can address this in the following ways. 1) The physical structure of as-measured defects (shown in Supplementary Figs. 4 and 12) are quite different, however all of the as-measured 1D defects show the same electronic structure and the capability to host a TLL. 2) We instead refer to these defects as 1D metals, due to similar electronic structure, and also highlight the lattice structure variations across different types of 1D defects that are measured.

We would like to highlight that the importance of the presented work is not centered on the structure of the 1D metallic defects or regarding the precise formation of MTBs. The main point we wish to convey is instead the ingredients required to form a TLL within a heterostructured material. The combined nARPES with STM/STS would show the important role graphene plays in order for this quantum phase transition to occur within a 2D material. The literature would suggest that both vacancy lines and MTBs would imbue metallic behavior to an otherwise semiconducting material. Thus, regardless if the defect is a vacancy line or an MTB, the conditions for a TLL to form are present and are represented throughout the results we present. The authors acknowledge that STM is a local measurement and other metallic 1D defect formations may occur, where we measure only, now, chalcogen vacancy lines, intermediate formations, and fully formed MTBs. However, additional metallic defects are possible. We also add the following text to help address the reviewers concerns.

Evidently, one key ingredient for the formation of a TLL in 2D material heterostructures is the nearest vertical contact. This motivated the detailed study of the local and macroscopic electronic structure to unveil the mechanism behind TLL formation in WS₂ 1DMs that are in contact with graphene.

We summarize the spectroscopic characteristics of chalcogen vacancy lines, strained MTB-like structures, and a fully formed MTB (formed across WS₂ edges) in Supplementary Fig. 12. All of these defects exhibit metallic characteristics within WS₂ and the capability for hosting a TLL, which contributes to the local modification of the electronic density of states. The significance of the underlying contact in the electronic structure formation is discussed in the next section.

Data obtained with ncAFM paired with STM/STS also confirms the formation of a TLL within WS₂ fully formed MTBs, intermediate MTB formations, and chalcogen vacancy lines.

There is not more I can say. I think the manuscript is not suitable for publication Nature Communications.

The authors would like to express their gratitude for the time spent in reviewing our work, especially as the authors missed an important vacancy formation in our earlier results that has since been refined from the last revision. We ask that the reviewer reconsider and would highlight the significant modifications made with the SPM side of the manuscript and overall results.

The authors would like to also note that in order to obtain the data to support the presence of vacancy lines and confirm electronic properties, multiple cycles of Ar bombardment and annealing were needed. After sufficient time, the substrate showed significant degradation. We believe this adds to the comprehensive study of 1D defect formations in WS₂.

Supplementary Fig. 3: **1DM Elongation and WS₂ Degradation.** Constant current image over WS₂ after (a) two SAA_{step} ($I_{tunnel} = 20$ pA, $V_{sample} = 1.2$ V), (b) 3 two SAA_{step} ($I_{tunnel} = 20$ pA, $V_{sample} = 1.4$ V), (c) four SAA_{step} ($I_{tunnel} = 25$ pA, $V_{sample} = 1.6$ V), and (d) five SAA_{step} ($I_{tunnel} = 25$ pA, $V_{sample} = 1.6$ V). Each cycle consists of a 30 second sputter and additional anneal. Scale bars, 4 nm. (e) dI/dV spectra recorded over non defective regions at each step and labeled by the total amount of sputter time ($V_{modulation} = 5$ mV). The expected E_{gap} of WS₂ decreases above four SAA_{step} and gains a metallic character, which is associated with material degradation.

Reviewer 4 (Remarks to the Author):

I have gone through the manuscript and the rebuttal letter for 2 rounds of review. It looks like the main debate is the atomic structure of the finite line observed by STM. Authors are arguing that they are symmetric inversion domain (SID), while they also observed some triangular shape of defects and they called that asymmetric inversion domain (AID). The original reviewer don't believe that the SID is MTB and consider it is just a line of vacancy complex.

Here is my opinion. From structural point of view, I totally agree with the reviewer. It is hard to imagine that a finite 1D line defect would be MTB, though the authors argue that it can be accommodated from defects at the end of the line. I have never seen such kind of structure and it would cause greater strain between the two sides since the orientation need to revert back 60 degree not only at the end of the line defect but also across the entire flake. So I agreed that the finite 1D line defects should not MTB, otherwise the authors should give clear atomic structure.

We greatly appreciate your detailed feedback and thoughtful comments on the structural aspects of the finite 1D line defects. We fully understand and acknowledge the concerns raised regarding the feasibility of these defects being fully formed mirror twin boundaries (MTBs), and we agree that such structures would be energetically unfavorable without relaxation mechanisms such as additional defects.

In response to these concerns, and following an extended investigation prompted in part by Reviewer 2's comments, we performed additional experiments and analyses over multiple samples. These revealed that all the defects we investigated (including chalcogen vacancy lines, 4I4P-like and 4I4E-like intermediate formations, fully formed MTBs, and other metallic defect structures) display remarkably similar electronic properties. We also include intermediate formations found in the literature^{1,2}. The similar electronic structure features found across numerous 1D defect symmetries has strengthened and broadened the scope

of our conclusions, as it demonstrates that the unique correlated Tomonaga-Luttinger Liquid (TLL) states are a general feature across all linear metallic defects observed in WS_2 when interfaced with graphene.

While we acknowledge the importance of understanding the precise atomic structure of these defects, we would like to reiterate that the primary focus of this manuscript lies in demonstrating the role of the graphene substrate in enabling correlated electronic states in WS_2 . The similarity in electronic properties across various defect types underscores the broader relevance of our conclusions and highlights the critical influence of graphene in this context.

We have adjusted the manuscript to reflect this generalization and have revised our terminology to describe the observed features as one-dimensional metallic defects (1DMs). This adjustment ensures that the scope of the manuscript remains focused on the role of graphene rather than the classification of specific defect symmetries.

Your feedback has been instrumental in refining this message, and we sincerely thank you for your efforts.

Supplementary Fig. 4: **Atomic Force Imaging.** (a) Constant current image over an isolated 4|4E-like 1DM defect ($I_{tunnel} = 30$ pA, $V_{sample} = 1.2$ V). Scale bar, 1.5 nm. (b) nAFM ($V_{sample} = 0.0$ V) collected over the same 4|4E-like 1DM with a ball model showing atomic locations. Scale bar, 0.4 nm. (c) A linescan over the red region shown in (b) that depicts two types of sulfur measured, where one is assigned to the top position (yellow) and the other to the bottom position (cyan) where metal sites appear as dips in frequency shift images. Depressions near the 4|4E-like 1DM reflect oxygen atom (circled in orange) chalcogen substitutions within an otherwise unmodified WS_2 lattice. A structure model is shown schematically in (d). A second isolated 1DM is shown in (e) that is identified to be an 4|4P-like 1DM ($I_{tunnel} = 30$ pA, $V_{sample} = 1.2$ V). Scale bar, 1.5 nm. (f) nAFM confirms the location of top sulfur, tungsten, and hollow sites ($V_{sample} = 0.0$ V), where a linescan across the 4|4P-like 1DM in (g) shows the location of dual tungsten surrounded by sulfur sites that is also shown schematically in (h). Scale bar, 0.4 nm. A 1DM chalcogen vacancy line is shown in (i) ($I_{tunnel} = 30$ pA, $V_{sample} = 1.2$ V). Scale bar, 1.5 nm. (j) Sulfur vacancies are measured in nAFM ($V_{sample} = 0.0$ V). A linescan across the 1DM in (k) shows the tungsten and lower sulfur sites, schematically depicted in (l). Scale bar, 0.4 nm.

Supplementary Fig. 5: **1DM Defect End Point Mapping.** (a) Constant current image over an isolated 1DM defect ($I_{tunnel} = 30$ pA, $V_{sample} = 1.2$ V) terminating in an end point defect. Scale bar, 1 nm. (b) Zoomed in image that is highlighted in (a) with a red box, where sequential (c) ncAFM with a CO functionalized tip ($V_{sample} = 0.0$ V) collected over the same 1DM showcase local structure. Scale bars, 1 nm. A chalcogen depletion region (blue) is located at the edge of the strained 1DM, which then forms into an isolated strained 4l4P-like structure (green). Chalcogen (sulfur) sites are highlighted with yellow spheres and metal (tungsten) sites are highlighted with green spheres. (d) Another defect is measured that appears as an intermediate formation before a fully relaxed MTB ($I_{tunnel} = 30$ pA, $V_{sample} = 1.2$ V). Scale bar, 1 nm. ncAFM over an endpoint region (highlighted with a red box) shows strained metallic-rich regions that form into a 4l4P-like structure.

Wang et al. Fig. S10: **The atomic structures of underlying S vacancy lines.** a, the atomic structure of three rows of S vacancy lines before the formation of Mo chains in Figure S9b. b, the atomic structure of inversion domain boundary before the formation of Mo chains in Figure S9c. c, the atomic structure of Mo₂S₃ before the formation of Mo chains in Figure S9e, f, i.

Lin et al. Fig. 2: **Agglomeration of selenium vacancies into line defects in monolayer MoSe₂.** (a) STEM Z-contrast image of the SL line defect with the DFT-optimized structure overlaid. Inset: Side view of the structure model. The nearby Se₂ columns are slightly misaligned. (b) STEM Z-contrast image of the 4|4E GB-like structure (highly strained 4|4E 60° GB) evolved from the SL line defect, which contains deformed strings of four-fold rings. The white dashed lines indicate the centers of these defects, and the yellow dashed arrows highlight the bond length of the Mo sublattice in the defect regions. Scale bars: 0.5 nm.

We make use of ncAFM with a functionalized CO tip to identify chalcogen vacancy lines, MTB-like defects, and chalcogen vacancy structures combined with MTB-like defects (see Supplementary Figs. 4 and 5). These defects are characterized by their distinct electronic behaviors—most are tungsten rich and exhibit electron-like characteristics, while sulfur rich IDMs demonstrate hole-like behavior (see Supplementary Fig. 6). This classification is based on the modulation of electron density with bias^{3,4}. Notably, the band dispersion observed in these defects mirrors that of known MTB structures^{5–7}. For the hole-like behavior, our findings align with the 4|4E MTB similar to what was reported by Jolie et al., and for the electron-like modulation, we observe this behavior in all tungsten rich defects that include chalcogen vacancy lines and symmetries resembling the 4|4P MTB found by Zhu et al. All structures display features of a TLL^{3,4,8}. In addition to defect structures measured with ncAFM, we also show a power-law dependence in as-acquired STM images in single-line defects and fully formed 60° MTB formations, where this behavior (characteristic of a TLL) is not present in structures that instead contact an underlying layer of WS₂ (see Supplementary Fig. 7). An isolated 1D MTB in an otherwise unaltered MX₂ lattice would be highly strained. If isolated MTB-like structures are formed, additional defects form (or combine) to help relax the lattice, such as Mo chains or chalcogen vacancies, at either the endpoints or within IDMs, as demonstrated in Supplementary Fig. 5. This observation highlights a defect at the IDM endpoint, where we measured a strained 4|4P-like symmetry thereafter. Our analysis reveals that among the types of IDM defects measured, the isolated MTB-like structures resemble intermediate defects in the process of forming fully relaxed MTB structures from V_S defects^{1,2,9}. Chalcogen vacancy lines, however, create less lattice strain of the isolated 1D defects measured with a single SAA_{step} cycle, and likely represent the majority of defects created.

However, I pretty appreciate the high quality data of the STM/STS and the nAPRES, and the conclusion is also interesting that worthy for publication. So I highly suggest the author to perform additional STEM/TEM experiments on the same sample to rule out the confusion of the atomic structures of the finite 1D defect. The worst scenario is that the author would find these finite 1D lines are not MTB but vacancy line, but it did not hurt the novelty of the work since no one have proposed that the 1D vacancy line can also host the TLL state. It make sense that it have been reported that the 1D vacancy line would also become metallic like the MTB. It also make sense that after annealing the existing S vacancy would reconstruct into either line defects or form inversion domain, so thiese two defects can co-exist. It is highly possible that the authors are observing the mixture of these two structures since they also see

difference in the STS between the SID and AID. Anyway I found the suggestion by the original reviewer is important and need to be clarified by additional experiments to make the atomic structure clear.

We deeply appreciate your recognition of the high-quality data presented in our manuscript. We wish to highlight the significance of our approach that combines nano angle-resolved photoemission spectroscopy (nARPES) with scanning tunneling microscopy/spectroscopy (STM/STS), which has enabled us to elucidate the critical role of graphene in facilitating the TLL emergence within WS₂ defects. This methodology allows for a unique insight into the formation of TLL within these heterostructures, which is a core achievement of our work. Given the controlled environment maintained during our experiments, we were able to directly compare data from nARPES to STM, ensuring consistency and reliability in our observations. This level of experimental control would not be easily replicated if incorporating an additional setup such as S/TEM, to which we currently lack access. Moreover, the dynamic nature of S/TEM measurements, with consistent electron irradiation, would induce changes in defect structures and could complicate the direct correlation of these findings with our current results.

The authors have instead performed multiple successive experiments to determine that vacancy lines do indeed form and, in light of this discussion in combination with new results, we have adjusted our introduction, results, and discussion in the manuscript to focus more on the one-dimensional nature of the defects rather than on their specific symmetry properties. We have increased the quality of the measured TLL formation (Fig. 2) above and below the measured E_{gap} , show dispersion formation in different defect lengths (Supplementary Fig. 8), and now directly compare measured electronic structure in fully formed MTBs, intermediate formations, and chalcogen vacancy lines (Supplementary Fig. 12).

We believe these clarifications and additions to our manuscript will further underscore the originality and significance of our work, and we are grateful for the opportunity to enhance our presentation based on your valuable feedback.

Fig. 2: Dense LDOS Mapping. (a) An electron-like 1DM is measured topographically, where the red line shown depicts a length of 18.96 nm ($I_{tunnel} = 20$ pA, $V_{sample} = 1.4$ V). Dense local density of states spectra is collected along (a) an electron-like 1DM ($1 \times 128 \times 400$ pixels) ($V_{modulation} = 5$ mV, $I_{set} = 150$ pA) beginning at the starred point along the red line. This is shown as a (b) function of bias and distance. The HOS and LUS are identified (c) at -0.032 V (orange) and 0.025 V (green). A 1D particle-in-a-box behavior is present above and below the HOS and LUS (maxima labeled as starred points). The number of nodes (N) increases linearly ($N+1$) from the LUS to LUS+1 (8 to 9), and also decreases from the HOS to HOS-1 (8 to 7). An FFT of (b) is shown in (d), where a spin and charge separation onset is seen above and below the E_F , and both the spin (blue) and charge (red) branches can be monitored from -0.15 V to 0.15 V. A K_c value of 0.47 is extracted from (d).

Supplementary Fig. 8: **Electron-like 1DM Dispersion Length Dependence.** Electron-like dispersions are collected on 1DMs with different lengths. Constant current images over 1DMs that are (a) 11.19 nm ($I_{tunnel} = 30$ pA, $V_{sample} = 1.2$ V), (b) 14.86 nm ($I_{tunnel} = 30$ pA, $V_{sample} = 1.4$ V), and (c) 20.44 nm ($I_{tunnel} = 80$ pA, $V_{sample} = 1.2$ V) are collected. Scale bars, 2 nm. Dense scanning tunneling spectra ($V_{modulation} = 5$ mV, $I_{set} = 150$ pA) are collected along the red line, beginning at the starred point. Results are shown for the (d) 11.19 nm defect (1x128x500 pixels), (e) 14.86 nm defect (1x128x400 pixels), and the (f) 20.44 nm defect (1x128x400 pixels). Using the HOS as a reference, a node spacing of is measured.

Supplementary Fig. 12: **Electronic Structure Comparison of Different 1DMs.** Constant current image over (a) a fully relaxed MTB across WS_2 edge regions ($I_{tunnel} = 30$ pA, $V_{sample} = 1.2$ V), (b) a vacancy line structure showing 120 degree rotation within WS_2 ($I_{tunnel} = 20$ pA, $V_{sample} = 1.4$ V), (c) a 4|4E-like structure ($I_{tunnel} = 30$ pA, $V_{sample} = 1.2$ V), and (d) a 4|4P-like 1DM ($I_{tunnel} = 30$ pA, $V_{sample} = 1.2$ V). Scale bars, 2 nm. (e) dI/dV spectra recorded over each 1DM defect case, where all show a measurable E_{gap} ($V_{modulation} = 5$ mV) near the E_F . In addition, we highlight dI/dV linescans extracted from the HOS as a function of defect distance (f-i), which all exhibit similar oscillatory behavior.

Evidently, one key ingredient for the formation of a TLL in 2D material heterostructures is the nearest vertical contact. This motivated the detailed study of the local and macroscopic electronic structure to unveil the mechanism behind TLL formation in WS_2 1DMs that are in contact with graphene.

We summarize the spectroscopic characteristics of chalcogen vacancy lines, strained MTB-like structures, and a fully formed MTB (formed across WS_2 edges) in Supplementary Fig. 12. All of these defects exhibit metallic characteristics within WS_2 and the capability for hosting a TLL, which contributes to the local modification of the electronic density of states. The significance of the underlying contact in the electronic structure formation is discussed in the next section.

Data obtained with ncAFM paired with STM/STS also confirms the formation of a TLL within WS_2 fully formed MTBs, intermediate MTB formations, and chalcogen vacancy lines.

The authors would like to also note that in order to obtain the data to support the presence of vacancy lines and confirm electronic properties, multiple cycles of Ar bombardment and annealing were needed. After sufficient time, the substrate showed significant degradation. We believe this adds to the comprehensive study of 1D defect formations in WS₂.

Supplementary Fig. 3: **IDM Elongation and WS₂ Degradation.** Constant current image over WS₂ after (a) two SAA_{step} ($I_{tunnel} = 20$ pA, $V_{sample} = 1.2$ V), (b) 3 two SAA_{step} ($I_{tunnel} = 20$ pA, $V_{sample} = 1.4$ V), (c) four SAA_{step} ($I_{tunnel} = 25$ pA, $V_{sample} = 1.6$ V), and (d) five SAA_{step} ($I_{tunnel} = 25$ pA, $V_{sample} = 1.6$ V). Each cycle consists of a 30 second sputter and additional anneal. Scale bars, 4 nm. (e) dI/dV spectra recorded over non defective regions at each step and labeled by the total amount of sputter time ($V_{modulation} = 5$ mV). The expected E_{gap} of WS₂ decreases above four SAA_{step} and gains a metallic character, which is associated with material degradation.

REFERENCES

- [1] Wang, X.-W. et al. Mass transport induced structural evolution and healing of sulfur vacancy lines and Mo chain in monolayer MoS₂. *Rare Met.* **41**, 333 (2022).
- [2] Lin, J., Pantelides, S. T., & Zhou, W. Vacancy-induced formation and growth of inversion domains in transition-metal dichalcogenide monolayer. *ACS Nano* **9**, 5189 (2015).
- [3] Jolie, W. et al. Tomonaga-Luttinger liquid in a box: Electrons confined within MoS₂ mirror-twin boundaries. *Phys. Rev. X* **9**, 011055 (2019).
- [4] Zhu, T. et al. Imaging gate-tunable Tomonaga-Luttinger liquids in 1H-MoSe₂ mirror twin boundaries. *Nat. Mater.* **21**, 748 (2022).
- [5] Batzill, M. Mirror twin grain boundaries in molybdenum dichalcogenides. *J. Condens. Matter Phys.* **30**, 493001 (2018).
- [6] Lehtinen, O. et al. Atomic scale microstructure and properties of Se-deficient two-dimensional MoSe₂. *ACS Nano* **9**, 3274 (2015).

- [7] Komsa, H.-P. & Krasheninnikov, A. V. Engineering the electronic properties of two-dimensional transition metal dichalcogenides by introducing mirror twin boundaries. *Adv. Electron. Mater.* **3**, 1600468 (2017).
- [8] Xia, Y. et al. Charge density modulation and the Luttinger Liquid state in MoSe₂ mirror twin boundaries. *ACS Nano* **14**, 10716 (2020).
- [9] Chen, Q. et al. Ultralong 1D vacancy channels for rapid atomic migration during 2D void formation in monolayer MoS₂. *ACS Nano* **12**, 7721 (2018).

Reviewer #2 (Remarks to the Author):

The authors now agree that some of the 1D structure could be vacancy lines. These are generally called 1D metals (1DM) in the manuscript. I think that is fine, although less satisfactory than actually knowing the defect type.

We thank the reviewer for their detailed and thoughtful comments. We have implemented all suggestions and carefully revised the manuscript and supplementary materials. Our changes aim to clarify the nomenclature and reinforce the physical interpretation of the defects observed, while preserving the core message of our work.

About the MTBness of the 1DMs, in the response the authors write: "Lin et al. does show an isolated 4|4E GB-like intermediate as shown above in Lin et al. Figure 2." This seems to be one of the main reasons for the confusion. They indeed seem to call a double vacancy line (as clearly depicted in the atomic structure inset) "4|4E GB-like structure" due to its similar chain of 4-rings in STEM images. However, it is obviously not a GB or MTB since the lattice orientation is the same around it. These structures only become GBs/MTBs in Figure 3(c) of that paper. In my opinion it is wrong to call the isolated versions of these MTB-like (since they are not mirror twins and not really even boundaries).

This is still reflected in the following parts of the manuscript. 1. In the text: "If isolated MTB-like structures are formed, additional defects form (or combine) to help relax the lattice, such as Mo chains or chalcogen vacancies, at either the endpoints or within 1DMs, as demonstrated in Supplementary Fig. 5." Again, there is no isolated MTB-like structures. You cannot have inverted lattice on the two sides of the 1DM just by adding some vacancies at the end points, because the inverted lattices cannot be smoothly connected around the end points. This requires connecting more MTBs to the end points, which makes them not isolated.

We agree with the reviewer that referring to isolated 1D vacancy line structures as "MTB-like" is not rigorous, as they do not exhibit mirrored lattice orientations and therefore cannot be classified as MTBs. We have removed the term "MTB-like" when referring to such isolated defects and now refer to them more precisely as strained 1DMs resembling 4|4P or 4|4E transitional configurations. Where necessary, we clarify that these structures may act as intermediates toward MTB formation but are not MTBs themselves, as shown in the following changes in the main text and in additional comments from the reviewer.

If isolated structures are formed, additional local strain-relieving features, such as Mo chains or chalcogen vacancies, can appear at either their endpoints or along their length, as demonstrated in Supplementary Fig. 5.

We make use of ncAFM with a functionalized CO tip to identify chalcogen vacancy lines, strained 1DMs resembling 4|4P or 4|4E intermediate defects, and combinations of chalcogen vacancies with strained 1DM defects (see Supplementary Figs. 4 and 5).

We summarize the spectroscopic characteristics of chalcogen vacancy lines, strained 1DM structures, and a fully formed MTB (formed across WS₂ edges) in Fig. 3.

2. Also in the text: "Our analysis reveals that among the types of 1DM defects measured, the isolated MTB-like structures resemble intermediate defects in the process of forming fully relaxed MTB structures from VS defects 28,31,36." Yes, they form from VS defects. No, there are not isolated MTB-like structures. Isolated vacancy-line structures are the intermediates as discussed in Ref. 31.

This sentence has been revised to: Our analysis reveals that among the types of 1DM defects measured, the isolated strained 1DM structures resemble intermediate defects in the process of forming fully relaxed

MTB structures from V_S defects¹⁻³. This new phrasing avoids suggesting that these structures are MTBs, and instead emphasizes their intermediate or strained character.

3. Supp. Figure 5(a-c), this is not "4|4-P like". (a) It seems like an isolated segment, and thus it cannot have inverted lattice on the two sides. (b) The images clearly show it is asymmetric perpendicular to the 1DM, while 4|4-P would be symmetric.

Thank you for your careful review. We have updated the text in Supplementary Fig. 5 to read as below:

Supplementary Fig. 5: **1DM Defect End Point Mapping.** (a) Constant current image over an isolated 1DM defect ($I_{tunnel} = 30$ pA, $V_{sample} = 1.2$ V) terminating in an end point defect. Scale bar, 1 nm. (b) Zoomed in image that is highlighted in (a) with a red box, where sequential (c) ncAFM with a CO functionalized tip ($V_{sample} = 0.0$ V) collected over the same 1DM showcase local structure. Scale bars, 1 nm. A chalcogen depletion region (blue) is located at the edge of the isolated and strained 1DM (further highlighted in green). Chalcogen (sulfur) sites are highlighted with yellow spheres and metal (tungsten) sites are highlighted with green spheres. (d) Another defect is measured that appears as an intermediate formation before a fully relaxed MTB ($I_{tunnel} = 30$ pA, $V_{sample} = 1.2$ V). Scale bar, 1nm. ncAFM over an endpoint region (highlighted with a red box) shows strained metallic-rich regions that form into a 4|4P intermediate structure.

4. Supp. Fig 3, these are all primarily vacancy lines as they are isolated segments. Also, in the caption "(b) 3 two SAA_{step} " should likely be "(b) three SAA_{step} ".

We appreciate the reviewer's careful observation and apologize for the typo. The caption for Supplementary Fig. 3 has been updated as below:

Supplementary Fig. 3: **IDM Elongation and WS₂ Degradation.** Constant current image over WS₂ after (a) two SAA_{step} ($I_{tunnel} = 20$ pA, $V_{sample} = 1.2$ V), (b) three SAA_{step} ($I_{tunnel} = 20$ pA, $V_{sample} = 1.4$ V), (c) four SAA_{step} ($I_{tunnel} = 25$ pA, $V_{sample} = 1.6$ V), and (d) five SAA_{step} ($I_{tunnel} = 25$ pA, $V_{sample} = 1.6$ V). Each cycle consists of a 30 second sputter and additional anneal. Scale bars, 4 nm. (e) dI/dV spectra recorded over non defective regions at each step and labeled by the total amount of sputter time ($V_{modulation} = 5$ mV). The expected E_{gap} of WS₂ decreases above four SAA_{step} and gains a metallic character, which is associated with material degradation.

5. Supp. Fig 4. These might indeed be as assigned, although a bit difficult to tell from the small figures.

We, again, thank the reviewer for their efforts. We also agree with the presented assignments and anticipate that similar methodologies of IDM defect formation in, perhaps, different TMD materials (such as WSe₂ as in Barja et al.⁴), that defect formation type may become easier to distinguish.

Other comments not related to MTB notation:

6. In the text: "These defects are characterized by their distinct electronic behaviors—most are tungsten rich and exhibit electron-like characteristics, while sulfur rich IDMs demonstrate hole-like behavior (see Supplementary Fig. 6)." One cannot directly deduce the electron- or hole-like behavior from the stoichiometry (S-/W-rich). In principle, vacancy lines, 4|4P and 4|4E MTBs are all W-rich. It is also not clear how the S/W-richness is deduced here as Supp. Fig. 6 only shows the electron-/hole-likeness.

We agree with the reviewer that albeit existing literature shows electron-like (hole-like) behavior in 4|4P (4|4E) MTBs, this may not be a definitive feature across all TMD materials. We have adjusted the text as below:

These defects are characterized by their distinct electronic behaviors—most exhibit electron-like characteristics, while other IDMs demonstrate hole-like behavior (see Supplementary Fig. 6). This classification is based on the modulation of electron density with bias^{5,6}. Notably, the band dispersion observed in these defects mirrors that of known MTB structures^{7–9}. For the hole-like behavior, our findings align with the 4|4E MTB similar to what was reported by Jolie et al., and for the electron-like modulation, we observe this behavior in chalcogen vacancy lines and strained IDMs resembling the 4|4P MTB found by Zhu et al.

7. "Dense STS is acquired over an anticipated electron-like IDM in Supplementary Fig. 16, where there is consistently no E_{gap} opening near the EF across the entirety of the defect." There are no visible IDM states in the few-layer WS₂ spectrum so how can you claim there is no E_{gap} opening?

We agree and have clarified the relevant sentence in both the main text. It now reads:

Dense STS is acquired over a IDM in Supplementary Fig. 15, where there are consistently no in-gap states, or evidence of a TLL, across the entirety of the defect.

8. Fig. 1(d), for the sake of completeness, it would be nice to see STS LDOS spectrum from a single vacancy on the multi-layer WS₂, if the authors happen to have one measured.

We appreciate this suggestion. However, as the key focus of our investigation is to unveil the role graphene plays in TLL formation within a IDM, the authors consider this an interesting area for future investigation that requires sufficient experimental evidence to properly address all types of isolated point vacancies that may form in multilayered WS₂ (such as top, bottom, divacancies, and neighboring effects (such as energetic peak shifts) of vacancies that may be induced within a single layer and across stacked layers).

In conclusion, I think I now understand where the confusion related to the MTB-like structures arises from, but in my opinion it is still wrong to call them MTB-like. Since "...the primary focus of this manuscript lies in demonstrating the role of the graphene substrate in enabling correlated electronic states in WS₂", it is probably fine to call these defects simply IDMs. However, it is then unfortunate that the evidence for TLL/correlated states is not very strong: spin-charge separation in Figure 2 is far from clear and the lack of any signal from the IDM on few-layer WS₂ makes it difficult to conclude much. For these reasons I would generally think that the manuscript does not meet the criteria for publication in Nature Communications, but at least I do not see major problems in it any more (after the authors have fixed the remaining issues mentioned above).

We appreciate the reviewer's fair assessment. While we respectfully acknowledge that the spin-charge separation evidence is subtle, we believe the combination of:

1. Power-law behavior in spatial LDOS decay,
2. $E_{\text{gap}} \propto 1/L$ scaling,
3. 1D particle-in-a-box nodal structure,
4. FT-STs showing split dispersions with extracted $K_c \sim 0.47$,

is in strong agreement with previous experimental benchmarks for TLLs. We have updated Figure 2 to better showcase the splitting of spin and charge with fitted lines for clarity and updated our methodology section.

Fig. 2: Dense LDOS Mapping. (a) An electron-like 1DM is measured topographically, where the red line shown depicts a length of 18.96 nm ($I_{tunnel} = 20$ pA, $V_{sample} = 1.4$ V). Dense LDOS spectra is collected along (a) an electron-like 1DM (1x128x400 pixels) ($V_{modulation} = 5$ mV, $I_{set} = 150$ pA) beginning at the starred point along the red line. This is shown as a (b) function of bias and distance. The HOS and LUS are identified (c) at -0.032 V (orange) and 0.025 V (green). A 1D particle-in-a-box behavior is present above and below the HOS and LUS (maxima labeled as starred points). The number of nodes (N) increases linearly ($N+1$) from the LUS to LUS+1 (8 to 9), and also decreases from the HOS to HOS-1 (8 to 7). An FFT of (b) is shown in (d), where a spin and charge separation onset is seen above and below the E_F , and both the spin (blue) and charge (red) branches can be monitored from -0.15 V to 0.15 V. A K_c value of 0.47 is extracted from (d).

FT-STs fit lines were derived from local extrema on spin and charge branches in accordance to analysis performed in Zhu et al⁶. A padding of 10 pixels was used to showcase the region of separation.

Reviewer #4 (Remarks to the Author):

I have gone through the rebuttal letter and the revised manuscript. I found the authors have addressed my comments and concerns nicely. The new data broadened the conclusion of TTL transition in 1D metallic defects including various structures like MTB, vacancy lines, etc., all of which presumably come from the underlying graphene substrate. I would like to recommend its publication. A minor point is that I suggest the author integrate the Fig. S12 in SI into the main text. This is very conclusive data showing the broadened claim of the revised manuscript.

Thank you very much for your review and we are very appreciative of your recommendation to publish in Nature Communications. We have moved Fig. S12 into the main text (now Fig. 3) as you suggest to help broaden the claim of TLL formation in different 1DMs within WS_2 .

REFERENCES

- [1] Wang, X.-W. et al. Mass transport induced structural evolution and healing of sulfur vacancy lines and Mo chain in monolayer MoS₂. *Rare Met.* **41**, 333 (2022).
- [2] Lin, J., Pantelides, S. T., & Zhou, W. Vacancy-induced formation and growth of inversion domains in transition-metal dichalcogenide monolayer. *ACS Nano* **9**, 5189 (2015).
- [3] Chen, Q. et al. Ultralong 1D vacancy channels for rapid atomic migration during 2D void formation in monolayer MoS₂. *ACS Nano* **12**, 7721 (2018).
- [4] Barja, S. et al. Charge density wave order in 1D mirror twin boundaries of single-layer MoSe₂. *Nat. Phys.* **12**, 751 (2016).
- [5] Jolie, W. et al. Tomonaga-Luttinger liquid in a box: Electrons confined within MoS₂ mirror-twin boundaries. *Phys. Rev. X* **9**, 011055 (2019).
- [6] Zhu, T. et al. Imaging gate-tunable Tomonaga-Luttinger liquids in 1H-MoSe₂ mirror twin boundaries. *Nat. Mater.* **21**, 748 (2022).
- [7] Batzill, M. Mirror twin grain boundaries in molybdenum dichalcogenides. *J. Condens. Matter Phys.* **30**, 493001 (2018).
- [8] Lehtinen, O. et al. Atomic scale microstructure and properties of Se-deficient two-dimensional MoSe₂. *ACS Nano* **9**, 3274 (2015).
- [9] Komsa, H.-P. & Krasheninnikov, A. V. Engineering the electronic properties of two-dimensional transition metal dichalcogenides by introducing mirror twin boundaries. *Adv. Electron. Mater.* **3**, 1600468 (2017).